# Motility precedes egress of malaria parasites from oocysts

**Dennis Klug\*, Friedrich Frischknecht\***

Integrative Parasitology, Center for Infectious Diseases, Heidelberg University Medical School, Heidelberg, Germany

**Abstract** Malaria is transmitted when an infected *Anopheles* mosquito deposits *Plasmodium* sporozoites in the skin during a bite. Sporozoites are formed within oocysts at the mosquito midgut wall and are released into the hemolymph, from where they invade the salivary glands and are subsequently transmitted to the vertebrate host. We found that a thrombospondin-repeat containing sporozoite-specific protein named thrombospondin-releated protein 1 (TRP1) is important for oocyst egress and salivary gland invasion, and hence for the transmission of malaria. We imaged the release of sporozoites from oocysts in situ, which was preceded by active motility. Parasites lacking TRP1 failed to migrate within oocysts and did not egress, suggesting that TRP1 is a vital component of the events that precede intra-oocyst motility and subsequently sporozoite egress and salivary gland invasion.

## Introduction

Many parasites switch between multiple hosts in order to complete their life cycles. These host switches are often accompanied by population bottlenecks where just a few parasites are sufficient for infection. For *Plasmodium*, the causative agent of malaria, a host switch is followed by an expansion of the parasite population in both the insect and the vertebrate host. A single *Plasmodium* parasite that establishes itself in the mosquito gut is enough to form an extracellular oocyst, in which hundreds of sporozoites can develop to colonize the salivary gland and be injected back into the vertebrate host. In *Plasmodium* species that infect mammals, a single sporozoite that successfully enters a hepatocyte is enough to produce thousands of progeny red-blood-cell-invading merozoites that then cause a full infection. In order to progress to the next developmental stage, the fully formed parasites need to escape their respective host cell or the oocyst.

These different immediate environments — a red blood cell within the blood stream, a hepatocyte within the liver parenchyma and an oocyst underneath the basal lamina of the mosquito gut — suggest that the different parasite stages use a mixture of unique and conserved processes for egress. There is evidence from all three stages to show that the release of parasites is dependent on a common set of specific proteins encoded by the parasite, including different proteases especially of the SERA family (*Arisue et al., 2007*; *Roiko and Carruthers, 2009*) but also other factors with no or unknown enzymatic activity (*Roiko and Carruthers, 2009*; *Wirth and Pradel, 2012*; *Talman et al., 2011*; *Ponzi et al., 2009*; *de Koning-Ward et al., 2008*; *Ishino et al., 2009*). Egress from blood cells and hepatocytes has been filmed in spectacular detail (*Abkarian et al., 2011*; *Sturm et al., 2006*), and exflagellation of microgametes has been studied extensively by light and electron microscopy (*Sinden et al., 1976*; *Deligianni et al., 2013*; *Wirth et al., 2014*; *Wilson et al., 2013*). These movies show, for example, the rapid rupture of the red blood cell membrane upon release of merozoites (*Abkarian et al., 2011*), as well as the perforation of the parasitophorous vacuolar membrane (PVM) and the erythrocyte membrane to enable exflagellation of activated male gametocytes (*Sinden et al., 1976*; *Deligianni et al., 2013*; *Wirth et al., 2014*). Finally, intravital

\*For correspondence: dennis.
klug@sciencebridge.net (DK);
freddy.frischknecht@med.uni-
heidelberg.de (FF)

**Competing interests:** The authors declare that no competing interests exist.

**eLife digest** Malaria is caused by a parasite transmitted by certain types of mosquito. The parasite lives in different organs within its vertebrate animal and insect hosts and to cope with these different environments it has a complex life cycle with several highly specialized life stages. To move from an infected mosquito into vertebrates the parasite produces spore-like cells called sporozoites that are able to enter different tissues and move very fast. These cells develop inside parasite-made structures called oocysts, which form at the stomach wall of the mosquito. After emerging from the oocyst, sporozoites float through the mosquito's circulatory system and eventually enter the salivary glands where they can be transmitted to vertebrates when the mosquito bites.

Efforts to develop malaria treatments and vaccines have focused on understanding the parasite's life cycle and identifying ways to control or eradicate key stages. Most researchers focus on the stage where the parasite is living in the vertebrate and actively causing disease, while the events in the mosquito are less intensely investigated. While several parasite proteins have been shown to be important for the release of sporozoites from oocysts, the molecular events leading to this release have not yet been fully resolved.

Klug and Frischknecht used time-lapse microscopy to film the release of the sporozoites of a malaria parasite known as *Plasmodium berghei*. The experiments show that the sporozoites can leave oocysts in several different ways. Furthermore, Klug and Frischknecht identified a new parasite protein named TRP1 that is essential for the sporozoites to leave oocysts and invade the salivary glands. Sporozoites lacking TRP1 were not able to move and they were unable to leave the oocyst or invade the salivary glands.

Klug and Frischknecht propose a new working model of the molecular events that govern sporozoite release in which TRP1 is required for sporozoites to move prior to their exit from oocysts. In the future, using the same techniques to analyze genetically modified parasites will help to reveal more details about sporozoite release.

microscopy in mice revealed the formation of merozoite-containing vesicles, termed merosomes, that bud from the infected hepatocyte (*Sturm et al., 2006*; *Baer et al., 2007*). By contrast, we have no live visual information about sporozoite egress from oocysts. Indeed, despite the adaptation of new dynamic imaging approaches in parasite biology (*Frischknecht, 2010*; *De Niz et al., 2017*; *Amino and Suzuki, 2014*), the only evidence to show how sporozoites egress from oocysts are images from electron microscopy. These show that sporozoites can appear in holes within the oocyst wall and the basal lamina that surrounds the oocysts (*Strome and Beaudoin, 1974*; *Meis et al., 1992*; *Sinden and Strong, 1978*), but also suggest that oocysts could rupture to release many parasites simultaneoulsly (*Meis et al., 1992*). Different species might use or prefer different ways to egress (*Orfano et al., 2016*).

A number of proteins have been identified as essential for sporozoite egress from oocysts of the human malaria parasite *P. falciparum* and of the rodent model malaria parasite *P. berghei* (*Table 1*). Some of the parasite lines that lack these proteins cannot exit the oocysts but can still migrate when mechanically released from oocysts whereas others cannot. Curiously, parasites lacking the thrombospondin-related anonymous protein (TRAP) also fail to undergo productive motility and salivary gland invasion, yet they have no defect in egress from oocysts (*Sultan et al., 1997*; *Münter et al., 2009*). Nevertheless, sporozoites that lack TRAP are still able to perform forms of unproductive movement, during which the parasite is attached at one focal spot to the substrate (*Münter et al., 2009*), which might be sufficient to force egress from oocysts. However, the number of non-related proteins that function in egress, and the lack of any clear interaction between these proteins, suggests that even more proteins are involved in the egress of sporozoites from oocysts. It was recently shown that the TRAP-family member MTRAP, previously thought to be important for red blood cell invasion by merozoites, is crucial for the egress of gametocytes from host cells (*Kehrer et al., 2016a*; *Bargieri et al., 2016*). We rationalized that similar proteins might also play a role in oocyst egress and searched for distantly related TRAP-like proteins. This revealed an as yet uncharacterized protein that carries a single thrombospondin repeat, one of the two adhesive

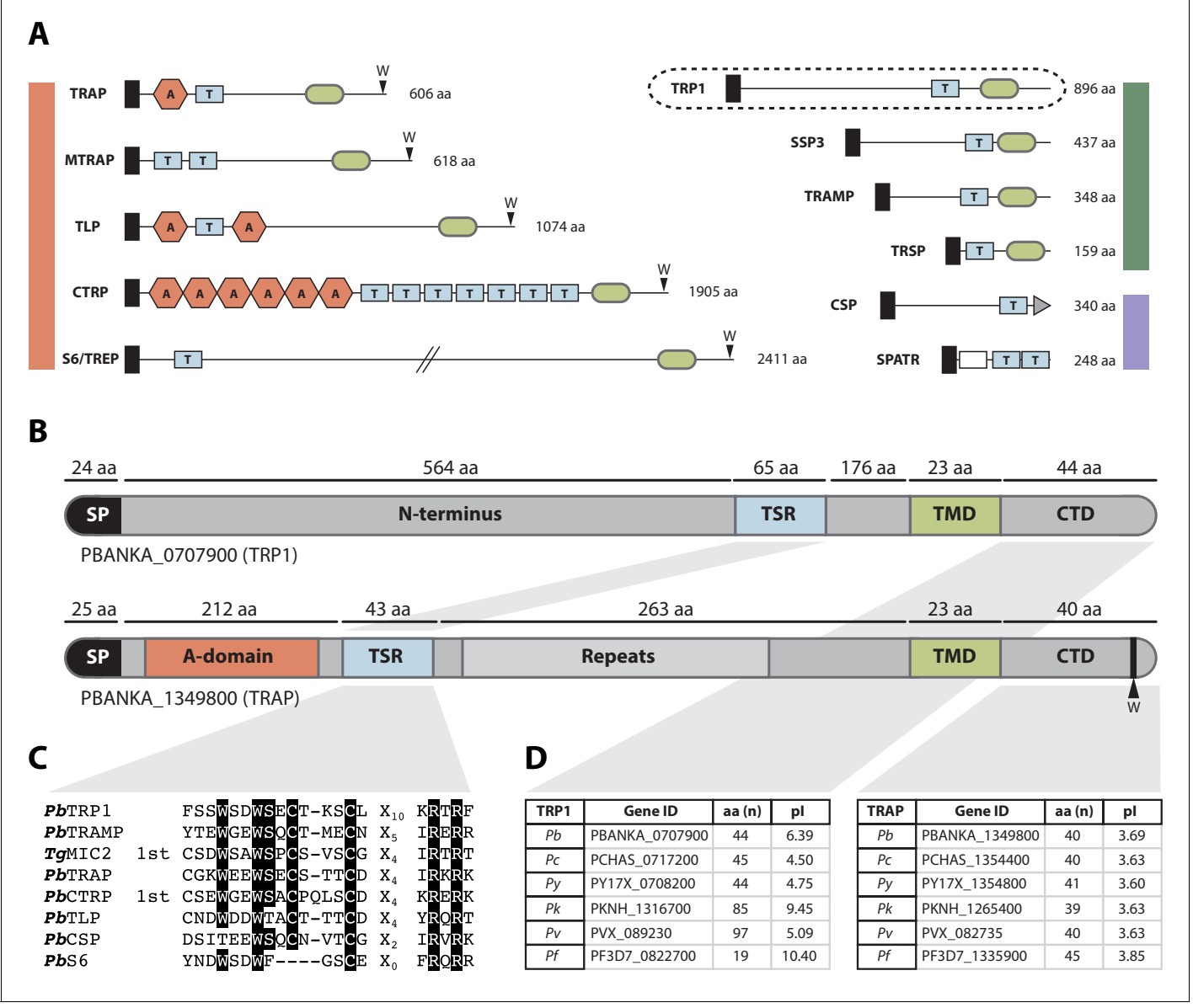

**Figure 1.** The thrombospondin-related protein 1 (TRP1) shares distinct domains with TRAP-family proteins, belongs to the family of TRAP-like proteins and is present in all *Plasmodium* species. (**A**) TSR-containing proteins in *Plasmodium*. TRAP-family proteins are marked with a red bar, whereas TRAP-related proteins are indicated by a green bar and other TSR-containing proteins by a blue bar. TRP1 is encircled by a dashed line (top right). Thrombospondin repeats are shown as blue boxes (labeled with T) and Von Willebrandt factor like A-domains are depicted as red hexagons (labeled with A). Signal peptides are shown as black boxes and transmembrane domains as light green ovals. CSP possesses a GPI-anchor (grey triangle), whereas SPATR harbors an EGF-domain (white box). Conserved tryptophans are indicated with a W. Protein schemes are not drawn to scale and amino acid numbers refer to *P. berghei* proteins. (**B**) Protein model of *Pb*TRP1 (PBANKA_0707900; 896 amino acids) in comparison to *Pb*TRAP (PBANKA_1349800, 606 amino acids, not to scale). Both proteins contain a signal peptide (SP), a thrombospondin type-I repeat (TSR), a transmembrane domain (TMD) and a cytoplasmic tail domain (CTD), but TRP1 lacks the conserved tryptophan (W) that is typically found at the C-terminus of TRAP-family proteins. Instead of the Von Willebrandt factor-like A-domain in TRAP, TRP1 contains a long N-terminal extension. (**C**) Multiple sequence alignment of the *Pb*TRP1 TSR with TSRs from the TRAP-family (*Pb*TRAP, *Tg*MIC2, *Pb*CTRP, *Pb*S6 and *Pb*TLP) and other TSR-containing proteins (*Pb*TRAMP and *Pb*CSP). (**D**) Length (in amino acids) and isoelectric point (pI) of the CTDs of TRP1 and TRAP from different *Plasmodium* species.

domains found in TRAP-family members. This thrombospondin-related protein 1, TRP1, also shared a transmembrane domain and a similar cytoplasmic tail with TRAP-family proteins (*Figure 1*). The generation of TRP1-deficient parasites revealed a function for TRP1 in sporozoite egress from

**Table 1.** Summary of known gene deletions and genetic modifications associated with defects in sporozoite egress from oocysts.

| Strain | Egress from oocysts | In vitro motility | Salivary gland invasion | Recognizable domain / function |
|---|---|---|---|---|
| wt | +++ | +++ | +++ | / |
| sera5(-)* | - | +++ | - | protease |
| csp-RIImut[27] | - | n.a. | n.a. | thrombospondin repeat (TSR) |
| csp(RI⁻)[57] | n.a. | +++ | ++ | TSR |
| csp(RII⁻)[57] | n.a. | - | + | TSR |
| ccp2(-) and ccp3(-) | - | ++ | n.a. | LCCL-like, ricin, discoidin, ApicA, levanase and neurexin-like domains |
| pcrmp3(-) and pcrmp4(-) | - | +++ | n.a. | CRM domain, EGF-like domain |
| gama(-)† | - | - | n.a. | / |
| siap-1(-) | + | - | + | / |
| orp1(-)‡ | - | +++ | - | histon-fold domain (HFD) |
| orp2(-)‡ | - | +++ | - | HFD |
| trp1(-) | - | +++ | - | TSR |

* Previously named ECP1 (**Aly and Matuschewski, 2005**).

† Previously named PSOP9 (**Ecker et al., 2008**).

‡ Information added during proof (**Currà et al., 2016**).

oocysts. Upon filming sporozoite egress from oocysts in isolated midguts, we found that motility precedes egress and that sporozoites that lack TRP1 are not motile within oocysts. Intriguingly, isolated trp1(-) sporozoites can undergo motility, suggesting that TRP1 might play a role in activating sporozoite motility in vivo prior to egress from oocysts.

## Results

### TRP1 belongs to the family of TRAP-related proteins

Within the mosquito, Plasmodium sporozoites display a repertoire of proteins that are important for egress from oocysts (SERA5, CSP, GAMA, SIAP-1, PCRMP3 and 4, and CCp2 and 3), motility (e.g. TRAP, MAEBL and S6) and salivary gland entry (e.g. TRAP and MAEBL). CSP in particular is interesting as it is essential for sporozoite formation (**Ménard et al., 1997**), as well as for efficient egress from oocysts once sporozoites have formed (**Wang et al., 2005**; **Coppi et al., 2011**) and also for sporozoite invasion of the salivary glands (**Coppi et al., 2011**). While the structure of CSP's N-terminus is unknown, the α-thrombospondin repeat (αTSR) at the C-terminal end as well as the repeat region between the N- and the C-terminus have been solved by x-ray crystallography and NMR (**Doud et al., 2012**; **Plassmeyer et al., 2009**; **Ghasparian et al., 2006**). The αTSR is especially interesting because this domain is believed to be involved in protein-protein interactions, as shown for the TSRs of thrombospondin-1 (**Iruela-Arispe et al., 1999**). Proteins containing TSRs are widespread among animals and protozoans and are mostly extracellular or secreted proteins (**Tucker, 2004**). Also, at least one TSR is contained within all proteins of the TRAP-family, including MTRAP, which was recently found to be important for gametocyte egress from red blood cells (**Kehrer et al., 2016a**; **Bargieri et al., 2016**; **Morahan et al., 2009**) (**Figure 1A**). In a search for uncharacterized TRAP-family-like TSR-containing proteins, we found a protein with unknown function in the rodent model parasite P. berghei (PBANKA_0707900). Because of the presence of a single TSR as a sole detectable domain, we will refer to this protein as thrombospondin-related protein 1 (TRP1).

The general domain composition of TRP1 shows many similarities to that of the thrombospondin-related anonymous protein (TRAP) (**Sultan et al., 1997**; **Morahan et al., 2009**) (**Figure 1B**). The TRP1 gene is intron-less, resides on chromosome 7 and encodes 896 amino acids. It can be found in all Plasmodium species and has a highly conserved core region that includes the TSR and the TMD

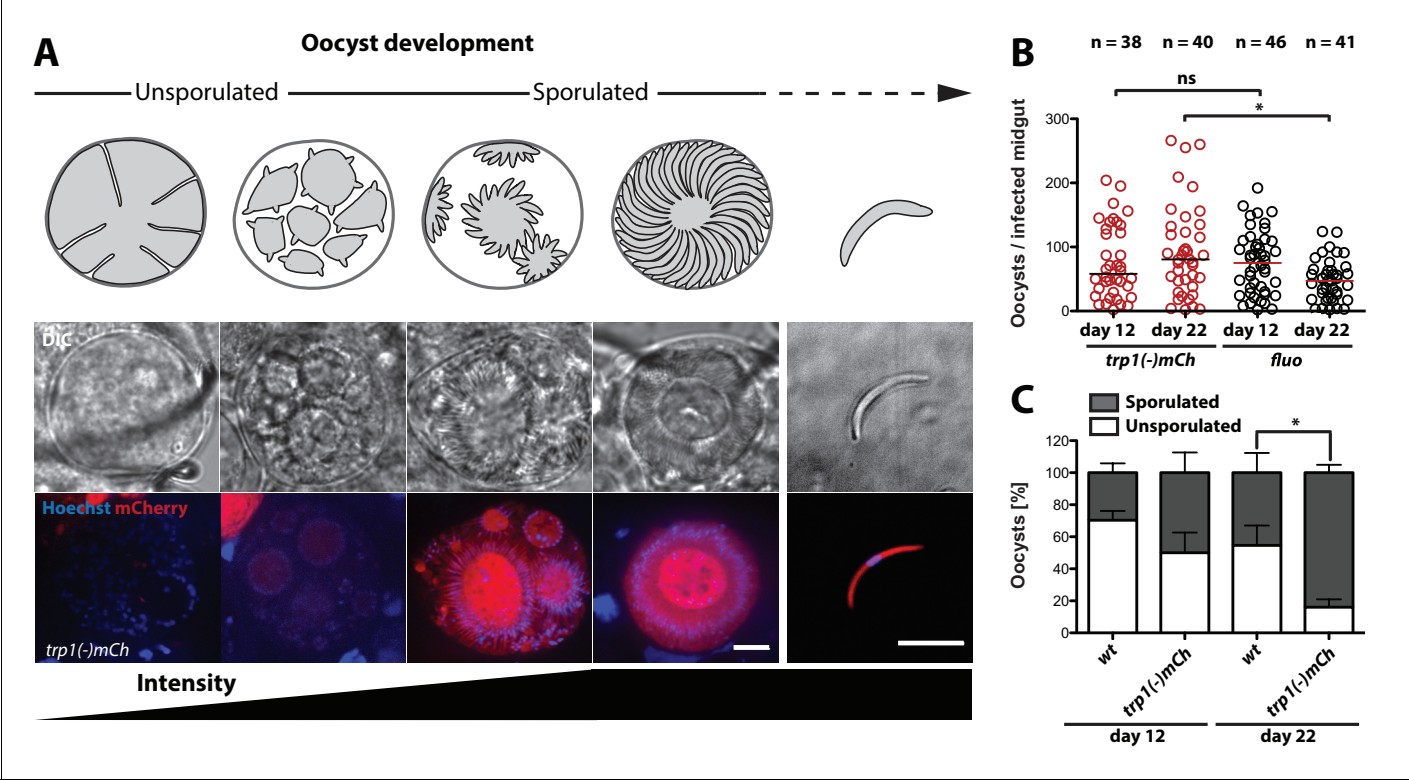

**Figure 2.** *trp1(-)* oocysts sporulate normally and persist in a sporulated state. (**A**) Expression of mCherry in *trp1(-)mCh* parasites was only observed in sporulating oocysts and sporozoites. The developmental stage of the oocysts is depicted schematically above the images, while the increase in fluorescence intensity is schematically indicated below. Strong mCherry expression was only observed in budding or mature oocysts. Scale bar: 10 μm. (**B**) Oocyst numbers of infected midguts for *trp1(-)mCh* and wild-type parasites at day 12 and day 22 post-infection. * depicts p<0.05; one-way ANOVA followed by a Kruskal-Wallis test. Horizontal bars indicate the median. Data were generated from two (*trp1(-)mCh*) and three (*fluo*) different feeding experiments, respectively. (**C**) Percentages of sporulated and unsporulated oocysts in *trp1(-)mCh* and wild-type infected midguts at 12 and 22 days post infection. * depicts p<0.05; one-tailed Student's t-test. The mean and the SEM are shown. Data were generated from three different feeding experiments.

The following figure supplements are available for figure 2:

**Figure supplement 1.** Generation and PCR analysis of *trp1(-)*, *trp1(-)mCh* and *trp1(-)rec* parasites.

**Figure supplement 2.** Classification of oocysts as unsporulated or sporulated.

**Figure supplement 3.** *trp1(-)* and *trp(-)mCh* midgut sporozoites are infective to mice if intravenously injected.

(Figure 5A, *Supplementary file 1*). Both TRP1 and TRAP contain a signal peptide (SP), a TSR, a transmembrane domain (TMD) and a cytoplasmic tail domain (CTD) (*Figure 1B*). However, TRP1 lacks the penultimate tryptophan (W) that has been shown to be important for TRAP function (*Kappe et al., 1999*) as well as the von Willebrandt factor like A-domain that is present at the N-terminus of TRAP and important for invasion (*Matuschewski et al., 2002*). Nevertheless, not all of the TRAP-family proteins — CTRP, S6/TREP/UOS3, TLP and MTRAP — possess an A-domain, but they are unified by the conserved C-terminal tryptophan residue (*Morahan et al., 2009*). In comparison to that of TRAP, the N-terminal domain of TRP1 is less conserved and varies widely in length between different *Plasmodium* species (332 aa in *P. vivax*; 651 aa in *P. falciparum*) (*Supplementary file 1*). A similar observation was made for the CTD. Whereas the CTD domain of TRP1 contains 19 amino acids in *P. falciparum*, it has a length of 97 amino acids in *P. knowlesi*. By contrast, the TSR of TRP1 is well conserved but has an unusual long insertion of 10 amino acids when compared to TSRs from other apicomplexan proteins (*Figure 1C*). Clusters of acidic amino

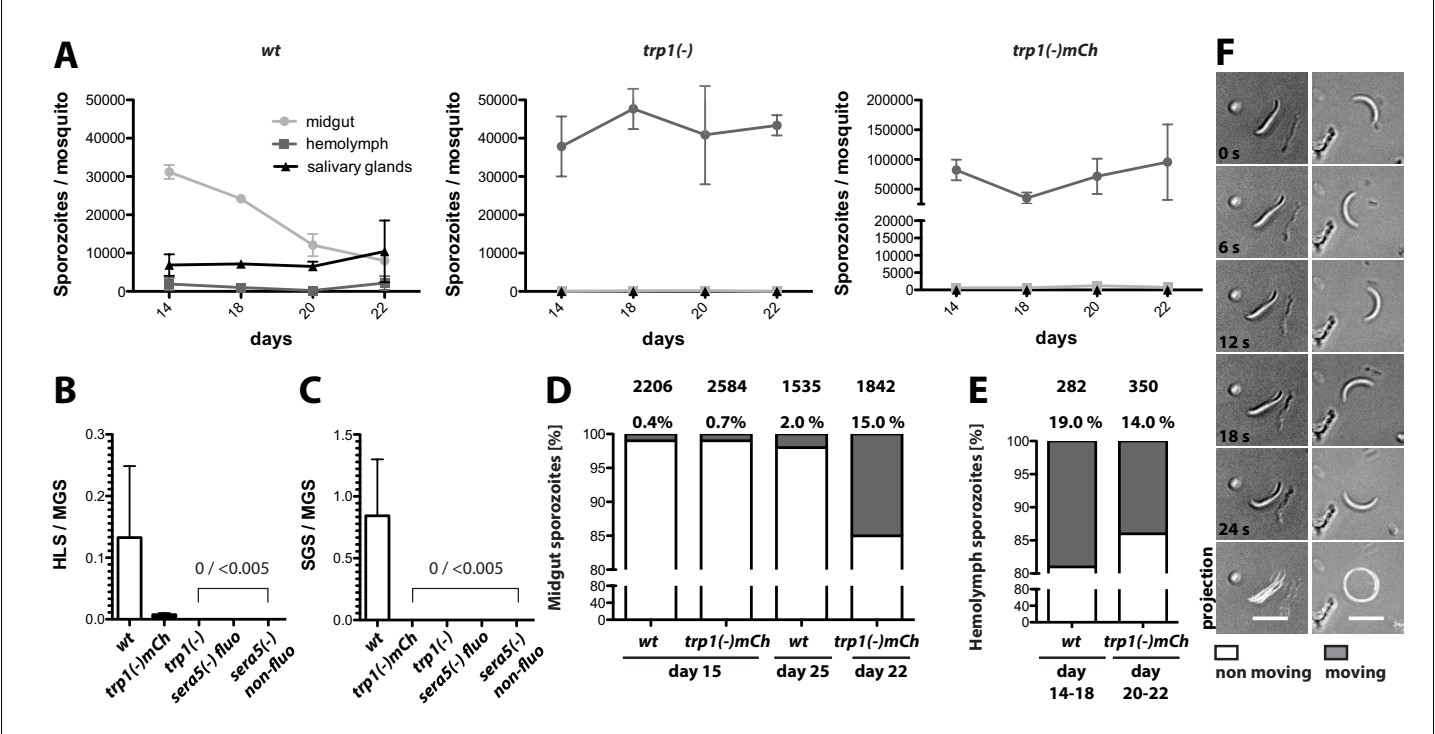

**Figure 3.** *trp1(-)* sporozoites are impaired in oocyst egress and salivary gland invasion but show normal gliding motility in vitro. (**A**) Numbers of wild-type, *trp1(-)* and *trp1(-)mCh* sporozoites in midguts, hemolymph and salivary glands over time. Shown are one to three countings per time point from one to three different feeding experiments. (**B**) Ratio of hemolymph (HLS) to midgut (MGS) sporozoites in wild-type-, *trp1(-)-* and *trp1(-)mCh*-infected mosquitoes. As negative control for a parasite that is not able to egress, a fluorescent and a non-fluorescent *sera5(-)* line were used. The bar represents the mean of four independent countings (ten mosquitoes each) at days 14, 17/18, 20 and 22 post infection of a selected feeding experiment. Error bars represent SEM. For absolute numbers see *Table 2*. (**C**) Ratio of salivary gland (SGS) to midgut (MGS) sporozoites corresponding to (**B**). The bar represents the mean, and error bars reflect SEM. For absolute numbers see *Table 2*. (**D**) Percentage of moving (dark) and non-moving (white) midgut sporozoites of wild-type and *trp1(-)mCh* at the indicated days post infection. Sporozoites were classified as moving if they were able to glide for at least one full circle within five minutes. All sporozoites that behaved differently were classified as non-moving. The number of investigated sporozoites is indicated on top of the bars. (**E**) Percentage of moving (dark) and non-moving (white) hemolymph sporozoites of wild-type and *trp1(-) mCh*. (**F**) Example of a non-moving (floating, left column) and a moving (circular movement, right column) *trp1(-)mCh* sporozoite isolated from the hemolymph. Scale bar: 10 μm.

The following figure supplement is available for figure 3:

**Figure supplement 1.** Generation and PCR analysis of *sera5(-) fluo* and *sera(5) non-fluo* parasites.

acids at the C-terminus have been shown to be important for TRAP function (*Kappe et al., 1999*). However, the calculated isoelectric point (pI) of the CTD of TRP1 varied widely between homologues from different *Plasmodium* species and indicated no enrichment for acidic amino acids (*Figure 1D*). Due to the overall similarity to TRAP-family proteins, we grouped TRP1 together with TRAMP (*Thompson et al., 2004*; *Siddiqui et al., 2013*), TRSP (*Labaied et al., 2007*) and SSP3 (*Harupa et al., 2014*) in the family of TRAP-related proteins that have a potential cytoplasmic tail domain (CTD) but lack the conserved tryptophan (*Figure 1A*). Homologues of TRP1 can be found in all *Plasmodium* species, but we could not identify homologues in other apicomplexans. However, homology predictions are difficult because TRP1 lacks a unique feature that allows unambiguous identification of protein homologues. As many TSR-containing proteins in other apicomplexan parasites are still uncharacterized, we cannot exclude the possibility that functional homologues exist outside of the genus *Plasmodium*.

**Table 2.** Sporozoite numbers in midgut (MG), hemolymph (HL) and salivary glands (SG). Sporozoites were counted at day 14, 17/18, 20 and 22 post infection. The mean and the standard deviation (± SD) of countings from two to three different feeding experiments are shown. Note that not all dissected mosquitoes were infected and hence numbers per infected mosquito are higher.

| Parasite line | No. of MG sporozoites per mosquito | No. of HL sporozoites per mosquito | MG / HL | No. of SG sporozoites per mosquito |
|---|---|---|---|---|
| wt | 18,100 (±10,600) | 1,400 (±1,700) | 13 | 7,800 (±5,300) |
| trp1(-) | 42,400 (±12,500) | 100 (±100) | 424 | 0 |
| trp1(-)mCh | 101,200 (±46,800) | 800 (±300) | 127 | 0 |
| sera5(-) fluo | 35,700 (±11,900) | 0 | / | 0 |
| sera5(-) non-fluo | 50,100 (±12,900) | 0 | / | 25 (±50) |
| gfp-trp1comp | 45,600 (±20,200) | 3,500 (±2,900) | 13 | 8,900 (±6,800) |
| gfp-trp1 | 9,800 (±10,200) | 1,500 (±2,100) | 7 | 1,900 (±2,000) |
| gfp-trp1ΔN | 18,900 (±12,700) | 1,000 (±700) | 19 | 0 |
| gfp-trp1ΔC | 52,700 (±9.500) | 1,100 (±300) | 48 | 0 |
| trp1-gfp parental | 3,400 (±1,600) | n.a. | n.a. | 175 (±100) |
| trp1-gfp clonal | 9,300 (±4,500) | 2,600 (±1,500) | 4 | 600 (±750) |

## Oocysts lacking TRP1 develop normally but sporozoites fail to egress

To study the function of TRP1, we created the knockout lines *trp1(-)* and *trp1(-)mCh* (*Figure 2—figure supplement 1*). *trp1(-)* is non-fluorescent, whereas in the *trp1(-)mCh* line, the *trp1* gene is replaced by the gene encoding the fluorescent protein mCherry. Expression of mCherry in *trp1(-)mCh* parasites was only observed in budding and completely sporulated oocysts up to hemolymph sporozoites (*Figure 2A*). To see if the lack of *trp1* affects the development of oocysts, we counted the number of oocysts at different time points after infection of mosquitoes by *trp1(-)mCh* or the fluorescent control line *fluo* (*Klug et al., 2016*), which expresses mCherry under control of the *CSP* promoter and GFP under control of the *ef1α* promoter (*Figure 2B*). Oocyst numbers for the control decreased between day 12 and day 22, whereas the number of oocysts in infected *trp1(-)mCh* mosquitoes remained stable. To screen for morphological differences between *trp1(-)mCh* and the control line, oocysts at day 12 and day 22 were also imaged and classified into oocysts containing mature or budding sporozoites (sporulated) and oocysts that didn't contain any sporozoite structures (unsporulated) (*Figure 2C*, *Figure 2—figure supplement 2*). This imaging showed that the percentage of *trp1(-)mCh* oocysts that had undergone or started sporogony (sporulated) was higher than the number of pre-mature (unsporulated) oocysts at both day 12 and day 22. While there was only a slight difference in this ratio between control and knockout on day 12, over 80% of the *trp1(-)mCh* oocysts were sporulated on day 22 compared with just 45% in the control line. As no morphological difference between sporulated knockout and control oocysts could be observed (*Figure 2—figure supplement 2*), we assumed that *trp1(-)mCh* sporozoites are fully developed but not able to egress. To study the infectivity of sporozoites lacking TRP1, we performed transmission experiments using either bites by infected mosquitoes or intravenous injections of 400,000–500,000 midgut sporozoites (Table 3, *Figure 2—figure supplement 3*, Figure 4—figure supplement 2). These experiments showed that mosquitoes infected with both knockout lines could not transmit the parasites to mice. However, we observed no difference in prepatencies between wild-type and *trp1(-)* parasites if midgut sporozoites were injected intravenously.

## TRP1-knockout parasites are impaired in oocyst egress and salivary gland invasion but show no defect in motility

Given the block in transmission by mosquitoes infected with the *trp1(-)* or *trp1(-)mCh* knockout parasites, we investigated the number of sporozoites in the midgut, hemolymph and salivary glands over time. In mosquitoes infected with wild-type parasites, midgut sporozoites start

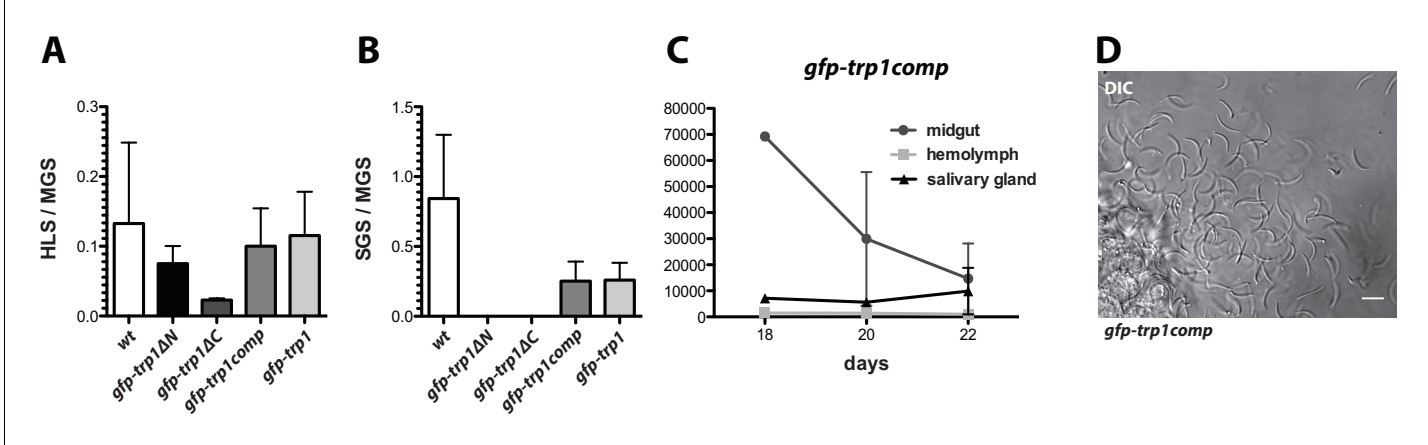

**Figure 4.** Complementation of *trp1(-)* parasites with full-length but not truncated TRP1 restores the wild-type phenotype. (**A**) Ratio of hemolymph sporozoites (HLS) to midgut sporozoites (MGS) and (**B**) of salivary gland sporozoites (SGS) to midgut sporozoites (MGS) for *gfp-trp1ΔN*, *gfp-trp1ΔC gfp-trp1comp* and *gfp-trp1* lines in comparison to wild-type (*wt*) parasites. The bar charts show the mean of four independent countings (10 mosquitoes each) at days 14, 18, 20 and 22 post infection of a selected feeding experiment. For absolute numbers see *Table 2*. Error bars represent SEM. (**C**) Sporozoites of *gfp-trp1comp* in midguts, salivary glands and hemolymph counted over time; 1–2 countings per timepoint. (**D**) Mechanically ruptured salivary gland releasing *gfp-trp1comp* sporozoites. Scale bar: 10 µm.

The following figure supplements are available for figure 4:

**Figure supplement 1.** Generation and PCR analysis of *gfp-trp1comp*, *gfp-trp1*, *gfp-trp1ΔN* and *gfp-trp1ΔC* parasites.
**Figure supplement 2.** TRP1 is essential for transmission by infected mosquitoes.

to egress from oocysts 11–12 days post infection; hence their numbers in the midgut decrease over time, while the numbers of hemolymph and salivary gland sporozoites slowly increase (*Figure 3A*). By contrast, mosquitoes that are infected with *trp1(-)* or *trp1(-)mCh* consistently showed high numbers of midgut sporozoites until 22 days post infection, while we observed only few sporozoites in the hemolymph and none in salivary glands (*Table 2*, *Figure 3A*). To visualize this effect more clearly, we calculated the ratio of hemolymph sporozoites (HLS) to midgut sporozoites (MGS) and salivary gland sporozoites (SGS) to MGS. The ratio for SGS to MGS was zero for both *trp1(-)* lines, while the ratio for HLS to MGS was either zero (*trp1(-)*) or very low (*trp1(-)mCh*) (*Figure 3B,C*). We also generated two *sera5(-)* strains (*Figure 3—figure supplement 1*) that were used as control for a non-egressing strain (*Aly and Matuschewski, 2005*). Ratios of SGS to MGS and HLS to MGS for both strains gave similar results as for *trp1(-)* and *trp1(-)mCh* (*Figure 3B,C*). To investigate the cause of failure to egress, we performed motility assays of wild-type and *trp1(-)mCh* midgut and hemolymph sporozoites. Both wild-type and *trp1(-)* sporozoites derived from midguts were able to perform the typical circular gliding motility of sporozoites (*Vanderberg, 1974*) but at very low rates (ca. 0.5–1% of the population) on day 15 (*Figure 3D*). This fraction increased in hemolymph sporozoites to 10–20% of the population, and again, no significant difference between *trp1(-)* and wild-type sporozoites was observed (*Figure 3E,F*). Interestingly the fraction of motile midgut sporozoites increased in the *trp1(-)mCh* line over time and reached a level comparable to that of wild-type hemolymph sporozoites at day 22 post infection (*Figure 3D,E*). By contrast, we observed no difference in the percentage of moving midgut wild-type sporozoites between day 15 and day 25 (*Figure 3D*).

## Complementation with full-length TRP1 restores infectivity

For the generation of the knockout lines *trp1(-)* and *trp1(-)mCh*, we made use of the positive-negative selection cassette *hdhfr-yfcu* (*Braks et al., 2006*). This made it possible to recycle the selection marker in *trp1(-)* parasites to generate the marker-free line *trp1(-)rec* (*Figure 4—figure supplement 1*). We used this line for complementation approaches with full-length as well as N- and C-terminally

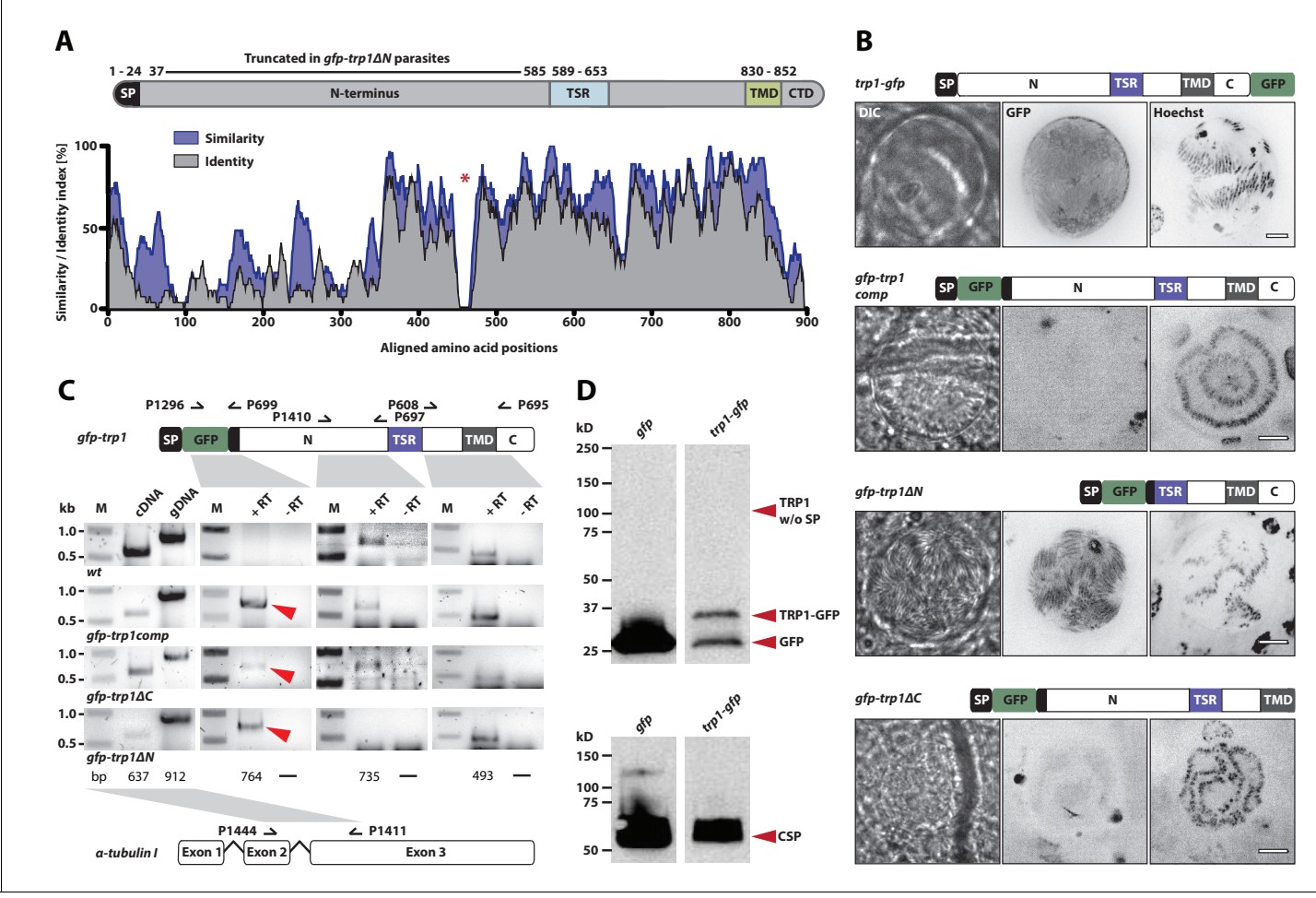

**Figure 5.** TRP1 is post-translationally processed. (A) Appearance of conserved residues (identity) and residues with similar charge (similarity) in *Pf*TRP1, *Pv*TRP1 and *Pk*TRP1 in reference to *Pb*TRP1. The graph corresponds to the protein model of TRP1 shown above. The gap marked by a red asterisk indicates an insertion in the sequence that is unique to *Pb*TRP1. Note the less conserved nature of part of the N-terminus. (B) Localization of TRP1-GFP, GFP-TRP1comp, GFP-TRP1ΔN and GFP-TRP1ΔC in oocsts 11–14 days post infection. Nuclear DNA is stained with Hoechst. Scale bar: 10 μm. See also **Video 1**. (C) RT-PCR of cDNA generated from midgut sporozoites. Purity of cDNA was tested with α-tubulin I primers amplifying a sequence from exon 2 to exon 3 (left). Splicing of the intron in-between the two exons resulted in a smaller PCR fragment compared to the gDNA. A *gfp:trp1* fusion transcript could be detected in *gfp-trp1comp*, *gfp-trp1ΔN*, and *gfp-trp1ΔC* (indicated by red arrowheads) but not in *wt* sporozoites. In addition, two PCRs were performed to detect two different parts of the *trp1* transcript. The gene and protein models shown below and above the images are not drawn to scale. (D) Western blot with 100,000 *trp1-gfp* and *csgfp* midgut sporozoites. The two lanes below show as loading control CSP, while the bands above show signals for GFP. Sporozoites expressing TRP1-GFP show a band at ~26 kDa that corresponds to free GFP and a second band at ~35 kDa that corresponds to GFP fused to the C-terminus and the transmembrane domain of TRP1. The predicted size of untagged TRP1 after cleavage of the signal peptide (~104 kDa) is indicated by a red arrow. Note that the shown images correspond to the same blot that was exposed for the same time. Lanes in-between the shown samples were only removed to simplify the representation.

The following figure supplement is available for figure 5:

**Figure supplement 1.** Generation and PCR analysis of parasites expressing TRP1 fused C-terminally to GFP (*trp1-gfp*).

truncated *trp1* mutants to further investigate the defects in oocyst egress and potential salivary gland invasion. The construct designed for complementation (*gfp-trp1comp*) contained the full-length *trp1* gene, whereas in the N-terminal deletion mutant (*gfp-trp1ΔN*), the sequence after the signal peptide until the start of the TSR (549 aa) was removed. On the other hand, the C-terminal deletion mutant (*gfp-trp1ΔC*) lacked the last 41 amino acids of the open reading frame, corresponding to the CTD (**Supplementary file 1**). All generated constructs contained an N-terminal GFP

**Table 3.** Infectivity of parasite lines to C57/BL6 mice. Data are shown for the TRP1-knockout lines *trp1 (-)* and *trp1(-)mCh* as well as for the TRP1 complementations *gfp-trp1comp*, *gfp-trp1ΔN* and *gfp-trp1ΔC* in comparison to wild-type (*wt – P. berghei* strain ANKA) and *gfp-trp1*. MG – midgut; i.v. — intravenous injection into tail vein.

| Parasite line | Route of inoculation | Mice positive #/# | Prepatency |
|---|---|---|---|
| *wt* | by bite | 4/4 | 3.25 |
| *wt* | 500,000 MG Sporozoites (i.v.) | 3/4 | 6.0 |
| *trp1(-)* clone 1 | by bite | 0/4 | ∞ |
| *trp1(-)* clone 3 | by bite | 0/4 | ∞ |
| *trp1(-)* clone 3 | 400,000 MG sporozoites (i.v.) | 2/4 | 6.0 |
| *trp1(-)mCh* | by bite | 0/4 | ∞ |
| *trp1(-)mCh* | 500,000 MG sporozoites (i.v.) | 4/4 | 6.5 |
| *gfp-trp1comp* | by bite | 4/4 | 3.0 |
| *gfp-trp1* | by bite | 4/4 | 3.5 |
| *gfp-trp1ΔN* | by bite | 0/4 | ∞ |
| *gfp-trp1ΔC* | by bite | 0/4 | ∞ |

placed in-between the signal sequence and the remaining ORF that allowed the visualization of the expression and localization of the fusion proteins. Transfections into *trp1(-)rec* gave rise to the three parasite lines *gfp-trp1comp*, *gfp-trp1ΔN* and *gfp-trp1ΔC* (*Figure 4—figure supplement 2*). The full-length construct was additionally transfected into *wt* to generate the line *gfp-trp1*. Both, *gfp-trp1comp* and *gfp-trp1* parasites showed normal ratios of HLS to MGS as well as of SGS to MGS that were comparable to *wt* (*Figure 4A,B*, *Table 2*). The observed sporozoites numbers in midgut, hemolymph and salivary glands between day 18 and day 22 in mosquitoes infected with *gfp-trp1comp* indicate the fully restored capacity to egress and invade (*Figure 4C,D*, *Table 2*).

Interestingly, we observed higher numbers of hemolymph sporozoites in mosquitoes infected with *gfp-trp1ΔN* and *gfp-trp1ΔC* than in those infected with the knockout strains. Despite similar numbers of hemolymph sporozoites in *gfp-trp1ΔN* and *gfp-trp1ΔC*, the ratio of HLS to MGS for *gfp-trp1ΔN* was more comparable to that for wild-type parasites, while the ratio in *gfp-trp1ΔC* was similar to that in *trp1(-)mCh* (*Figure 4A*, *Table 2*). This suggests that *gfp-trp1ΔN* sporozoites are more capable of egressing from oocysts than *gfp-trp1ΔC* sporozoites. By contrast, the ratio of SGS to MGS was zero for both *gfp-trp1ΔC* and *gfp-trp1ΔN*, indicating that the knockout phenotype is only partially restored in *gfp-trp1ΔN* parasites. The infectivity of *gfp-trp1comp*, *gfp-trp1*, *gfp-trp1ΔC* and *gfp-trp1ΔN* was also tested by natural transmission experiments involving mosquito bites (*Table 3*, *Figure 4—figure supplement 2*). While *gfp-trp1comp* and *gfp-trp1* showed normal infectivity in mice, no infection could be observed for *gfp-trp1ΔN* and *gfp-trp1ΔC*. These results matched the data for sporozoite numbers in the salivary glands that were zero for both mutants (*Table 2*). These data suggest that the N-terminus of TRP1 is important for oocyst egress and that both the N- and the C-terminus are essential for salivary gland invasion. Intriguingly, the N-terminus of TRP1 is the least conserved part of the protein across *Plasmodium* species (*Figure 5A*).

## TRP1 is post-translationally processed

The complemented lines *gfp-trp1comp*, *gfp-trp1ΔN* and *gfp-trp1ΔC,* as well as the line *gfp-trp1*, were tagged N-terminally with GFP to access the expression and localization of the fusion proteins in vivo. We further constructed a parasite line in which GFP was C-terminally fused to TRP1, termed *trp1-gfp* (*Figure 5—figure supplement 1*). This line showed no defect in sporozoite egress from oocysts but salivary glands contained consistently less sporozoites compared to wild-type (*Table 2*). Interestingly, we could only detect strong GFP fluorescence in the *trp1-gfp* line and weak GFP fluorescence in *gfp-trp1ΔN* parasites, while all other lines were non-fluorescent (*Figure 5B*). In contrast to *trp1(-)mCh* parasites, which express mCherry early during oocyst formation (*Figure 1A*), GFP expression in *gfp-trp1ΔN* parasites was only observed in oocysts with already matured sporozoites.

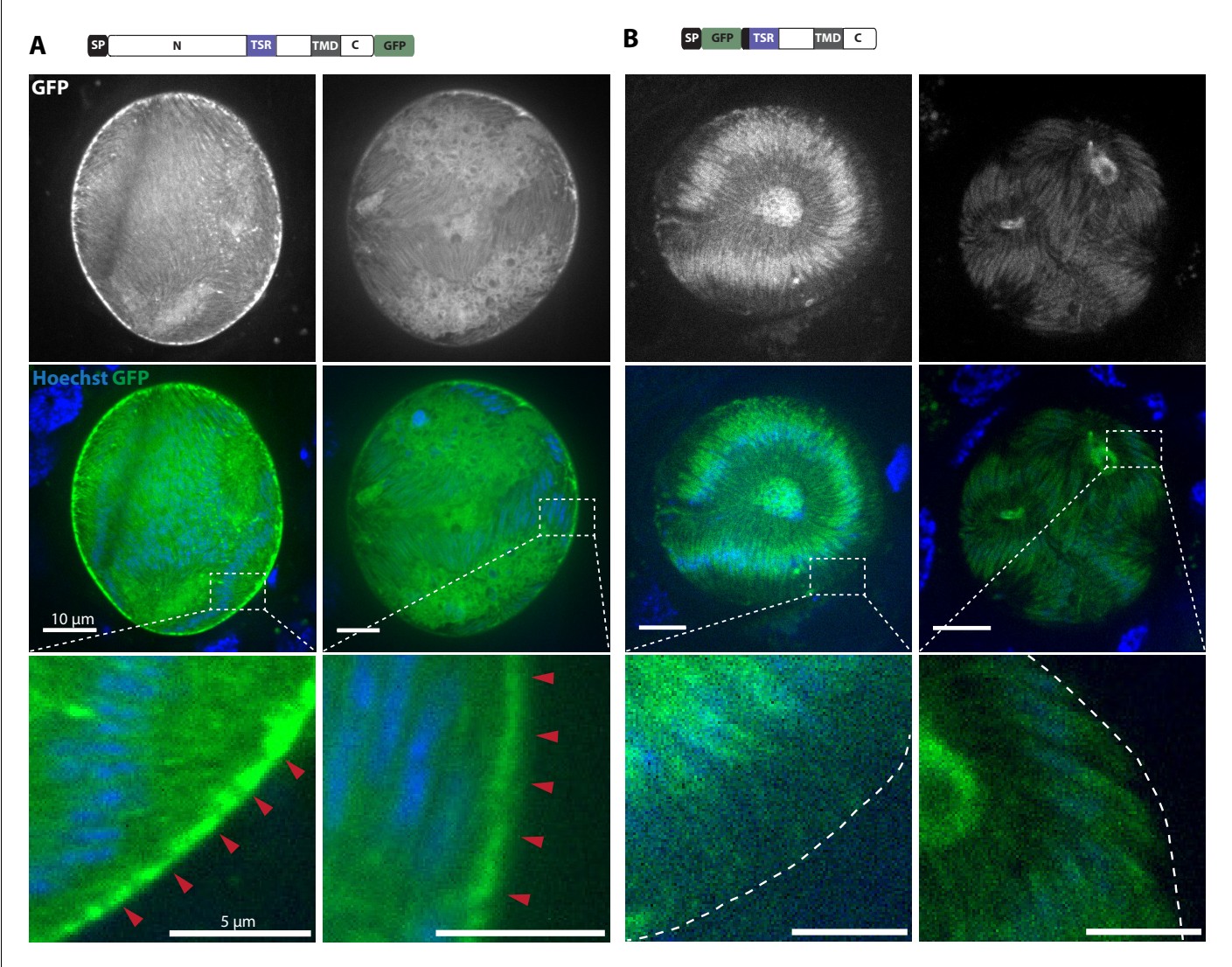

**Figure 6.** TRP1-GFP localizes to the oocyst wall while GFP-TRP1ΔN accumulates in the endoplasmatic reticulum (ER). (**A**) Localization of TRP1-GFP in oocysts 11–14 days post infection. Nuclear DNA is stained with Hoechst. The accumulation of GFP at the oocyst wall is indicated by red arrows in the zoomed images. See also **Video 1**. (**B**) Localization of GFP-TRP1ΔN at 11–14 days post infection. Nuclear DNA is stained with Hoechst. The dashed white line in the zoomed images indicates the oocyst wall.

This indicates that, beside the transcriptional regulation, TRP1 expression might also be post-transcriptionally regulated by the native 3'UTR, which is not present in *trp1(-)mCh* parasites. To test whether these lines transcribe the *gfp* and *trp1* genes as one transcript, we first performed RT-PCRs with cDNA generated from midgut sporozoites. We were able to amplify a *gfp:trp1* transcript in *gfp-trp1comp*, *gfp-trp1ΔN* and *gfp-trp1ΔC* sporozoites while no transcript could be detected in *wt*. In addition, we performed two RT-PCRs for *trp1* transcripts, amplifying sequences encoding the TSR and the N-terminus. While the PCR amplifying the TSR sequence gave a product in all lines including *wt*, a product for the N-terminal PCR was only observed in *gfp-trp1comp*, *gfp-trp1ΔC* and *wt* parasites but, as expected, not in *gfp-trp1ΔN* parasites (**Figure 5C**). Finally, we performed Western blotting, which revealed only a short GFP-fusion protein as well as GFP for the C-terminally tagged full-length TRP1 (**Figure 5D**). We could not detect full-length GFP-TRP1, suggesting that the protein is cleaved, and also failed to detect protein for the weakly GFP-expressing N-terminal deletion parasites.

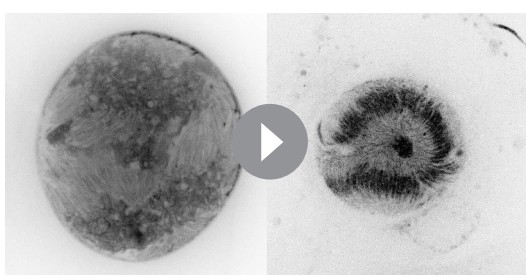

**Video 1.** Sections in Z-direction through oocysts expressing TRP1-GFP and GFP-TRP1ΔN. Movie showing slices in Z-direction of an oocyst expressing TRP1-GFP (left) and an oocyst expressing GFP-TRP1ΔN (right). Oocysts were imaged with a spinning disc confocal microscope (Nikon Ti series) and a 60x objective (CFI Apo TIRF 60x H; NA 1.49). Only the GFP signal is shown.

Curiously, the C-terminal GFP-fusion protein of the full-length TRP1 localized strongly at the periphery of the oocyst (*Figure 6A*, *Video 1*). By contrast, the GFP-fusion protein of the N-terminally deleted TRP1 did not show this localization and appeared to accumulate only within the sporozoites (*Figure 6B*, *Video 1*). Investigation of single sporozoites showed that the C-terminally tagged TRP1 localized at the periphery towards the rear end of the sporozoite, as well as within the sporozoite at the apical end (*Figure 7A*, *Video 2*). By contrast, the GFP signal in the *gfp-trp1ΔN* sporozoites localized to membranes within the parasite, presumably the endoplasmatic reticulum (ER) as indicated by accumulations around the nucleus (*Figure 7—figure supplement 1*). A parasite line expressing cytoplasmic GFP was used as a control (*Figure 7C*). By using anti-GFP antibodies in fixed sporozoites, we were not able to detect GFP-TRP1ΔN on the sporozoite surface (*Figure 7—figure supplement 1*). Comparison of the localizations of TRP1-GFP and GFP-TRAP (*Kehrer et al., 2016b*) showed that the two proteins were localized differently (*Figure 8*). While GFP-TRAP appears mostly localized to micronemes at the front end of the sporozoite, TRP1-GFP appears to localize in what might be a subset of micronemes or a different organelle that does not extend all the way to the front. These data together indicate that TRP1 is transported through the ER, is post-translationally processed and can accumulate at the periphery of the parasite on its rear end.

## Synchronous activation of mature sporozoites is crucial for effective egress from oocysts

Even if a low number of hemolymph sporozoites could be detected in the *trp1(-)* and *trp1(-)mCh* knockout lines, the presented results indicate that the initial function of TRP1 lies in oocyst egress. To probe this in more detail, we established two new assays to image sporozoite egress from oocysts. These assays imaged extracted midguts placed either on microscope slides and pushed down by a coverslip or in a non-compacted setting in glass-bottom Petri-dishes (*Figure 9A,B*). Individual midguts were placed in insect medium (Grace's medium, Gibco™, Thermo Fisher Scientific, Waltham, MA) and imaged either at high or low magnification for up to one hour. As expected, the frequency of egress events was low for both setups (*Figure 9A,B*). However, we were able to observe dozens of egress events. In one type of egress, mostly observed in the slide-coverslip setup, sporozoites were seen to be moving actively inside oocysts followed by egress from oocysts in a manner resembling the merosomes formed by late liver stages (*Sturm et al., 2006*; *Baer et al., 2007*) (*Figure 9C*, *Video 3*). We thus termed these vesicle-like deformations of the cyst wall sporosomes. From a total of over 800 imaged oocysts, we observed active sporozoite motility in 5–6% and egress-like events in about 3% of the wild-type oocysts. The same type of events with similar frequencies were also observed at the 2015 and 2016 *Biology of Parasitism* courses at Woods Hole using mosquitoes from two different insectaries. All egress-like events were preceded by sporozoite motility. By contrast, we observed no egress-like events in the *trp1(-)mCh* line and also no actively moving sporozoites within oocysts (*Figure 9A*). Although no egress could be observed in the knockout line, we were concerned that the pressure on the midguts induced by the overlaying cover slip might force sporozoite egress. Therefore, we performed the same assay in glass-bottom Petri-dishes, in which midguts were simply placed in medium with no lid. To this end, we used a fluorescent line (*fluo*) because imaging was performed with low magnification (10x) and egress events were only visible with fluorescence detection. In these experiments, we were able to observe motile sporozoites and their egress in the control at rates similar to those seen in the previous assay. Furthermore, we did not detect motility or egress in the TRP1 knockout (*Figure 9B*, *Videos 4–6* and *Video 7*). In this assay, we observed a variety of events like rapid bursting of oocysts,

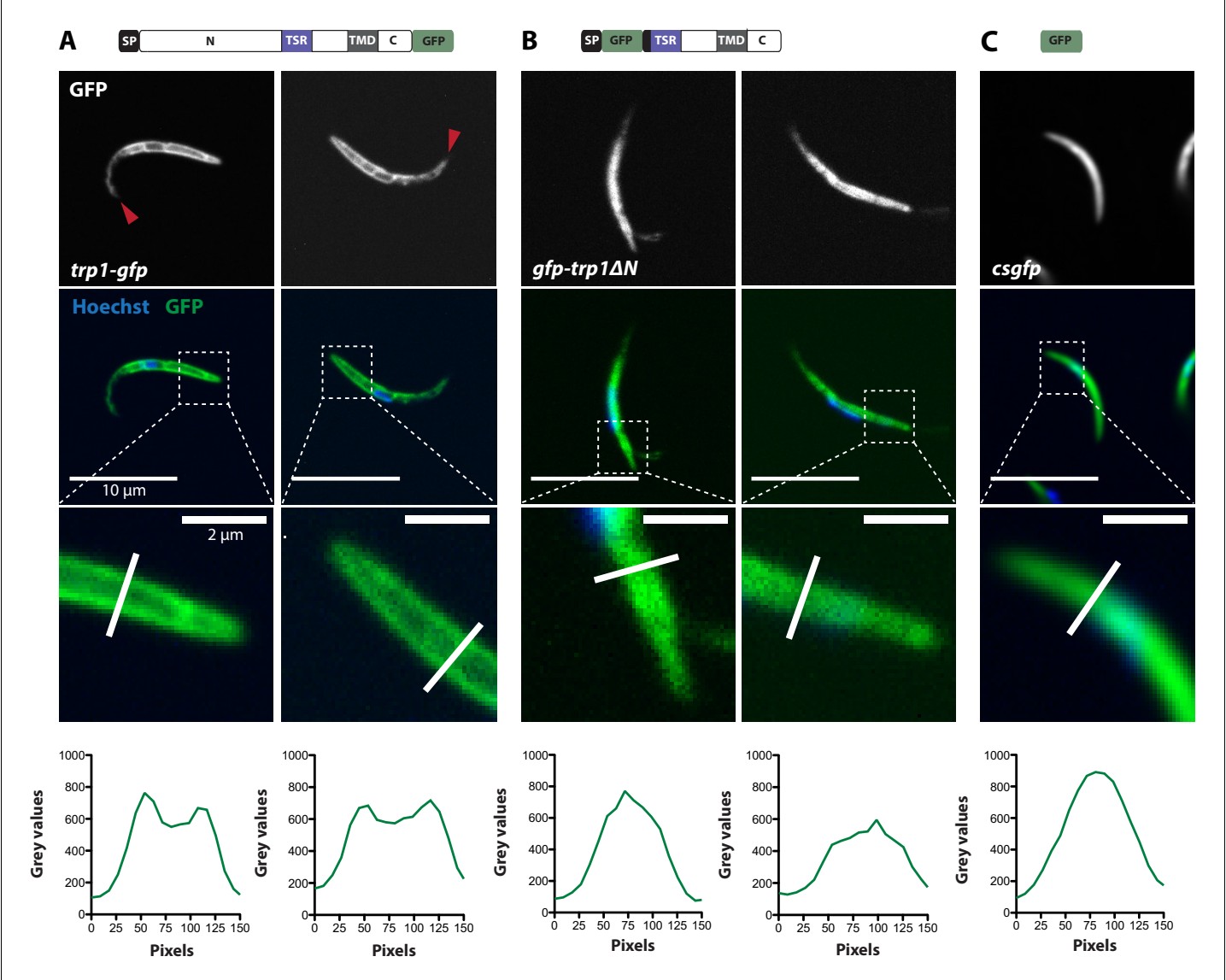

**Figure 7.** TRP1-GFP but not GFP-TRP1ΔN localizes in a polarized fashion at the sporozoite periphery. (**A**) Live imaging of hemolymph sporozoites expressing TRP1-GFP. The line plot below shows the intensity of grey values along the white line indicated in the zoomed image, showing the proximal end of the sporozoite. The GFP signal localizes close to the plasma membrane indicated by the intensity profile showing two peaks on both sides of the sporozoites. The red arrows point to the apical tip of the sporozoites. See also *Video 2*. (**B**) Live imaging of midgut sporozoites expressing GFP-TRP1ΔN. The GFP signal is not equally distributed as seen in control parasites in (**C**) but does not localize close to the plasma membrane as shown in (**A**). (**C**) Live imaging of a salivary gland sporozoite expressing cytoplasmic GFP. In contrast to (**A**) and (**B**), the GFP signal is equally distributed within the cytoplasm.

The following figure supplement is available for figure 7:

**Figure supplement 1.** GFP-TRP1ΔN does not localize on the sporozoite surface.

sporozoites budding from oocysts and sporozoites moving inside oocysts or their surrounding tissue (*Videos 4–6*). In parasites lacking the protease SERA5, we confirmed sporozoite motility within oocysts as described previously (*Aly and Matuschewski, 2005*) and did also not witness any egress (*Figure 9A,B* and *Videos 7* and *8*). Intriguingly, *sera5(-)* parasites showed about four and eight times more oocysts with motile sporozoites than wild-type oocysts in both assays (*Figure 9A,B*),

respectively. Considering that *sera5(-)* sporozoites do not exit oocysts, this further suggests that motility precedes egress.

## Discussion

### Visualizing sporozoite egress: bursting and budding within vesicle-like structures

Here, we described for the first time the imaging of *Plasmodium* sporozoite egress from oocysts in situ, which revealed a number of different types of egress events. The most striking was the apparent bursting of sporozoites and the apparent budding of sporozoites within vesicle-like deformations of the cyst wall that we termed sporosomes (*Figure 9C,D*; *Videos 3* and *7*). We imaged 547 control oocysts over a total of 16 hr and observed egress events in about 3% of these oocysts, suggesting a rate of 0.5–1 egress events per hour. At this rate, 50% of oocysts in a medium infected midgut (~100 oocysts) would be emptied in 2–3 days. These numbers suggest that the observed events might well reflect those occurring in vivo. Naturally, in vivo observations would be desirable but intravital imaging in mosquitoes is challenging due to the opaque nature of the chitinous exoskeleton. This can be partially overcome by blood feeding mosquitoes just before imaging; this causes the abdomen to inflate to such a degree that individual oocysts can be imaged by fluorescence

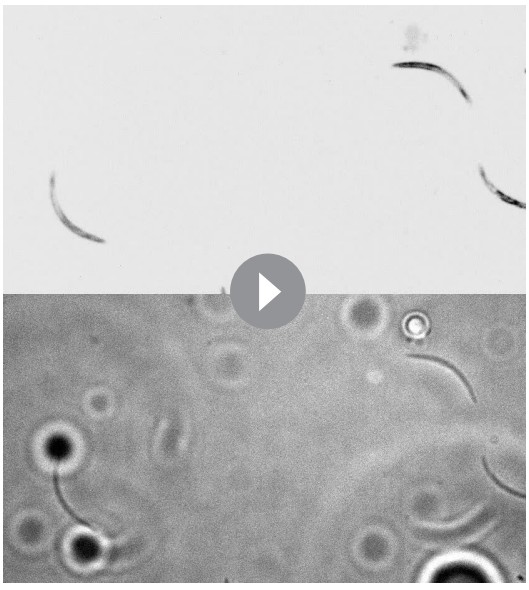

**Video 2.** Salivary gland sporozoites expressing TRP1-GFP gliding. Movie showing salivary gland sporozoites expressing TRP1-GFP gliding close to a salivary gland imaged on a spinning disc confocal microscope (Nikon Ti series) with a 60x objective (CFI Apo TIRF 60x H; NA 1.49). The GFP signal and the differential interference contrast (DIC) are shown beside each other. Note the intra-sporozoite movement of the TRP1-GFP signal, the peripheral localization is mainly at the rear end of the motile sporozoites. Time between frames: 1 s.

microscopy. Yet, it is unclear whether a recent blood meal would influence egress. The application of fiber optic imaging could overcome this challenge (*Sum and Ward, 2009*). In this methodology, a thin fiber is introduced into the mosquito and individual oocysts should be visible to the patient observer. However, the low rates of egress events would make imaging egress events in vivo a formidable challenge.

The observation of sporozoite budding into what appears to be membrane-delimited vesicles was both unexpected and curious, as the origin of the surrounding 'membrane' is not obvious. The plasma membrane of the ookinete develops into the plasma membrane of the developing oocyst, which in turn forms the plasma membrane of the sporozoites (*Vanderberg and Rhodin, 1967*; *Thathy et al., 2002*). Hence, sporozoites differ from intracellular growing stages that are surrounded by a parasitophorous vacuole and a host cell plasma membrane: there should be no membrane around the formed sporozoites since they are only surrounded by the oocyst wall. Yet, the appearance of these sporosomes suggests that some form of delimiting 'membrane' is present. Earlier scanning EM images of infected midguts suggested that oocysts can bud off smaller cysts that were called satellites (*Strome and Beaudoin, 1974*). It is not clear how satellite formation can occur or how it is initiated, but these observations suggest that the oocyst wall might be not completely rigid and could show some level of elasticity. Hence part of the delimiting 'membrane' of the sporosomes could be derived simply from the oocyst wall and thus not be a lipid membrane but a thin sheet of oocyst wall material, similar to that which might surround sporozoites within satellites. As we only observed sporosome formation at midguts infected with wild-type parasites, it might well be that the oocyst wall is thinned by a succession of parasite-initiated proteolytic events, which are not initiated in *trp1(-)* or *sera5(-)* parasites. These events likely destabilize the wall in a way that enables sporozoites to egress via rupture or budding. Yet, EM imaging of sporulated *trp1(-)mCh* and

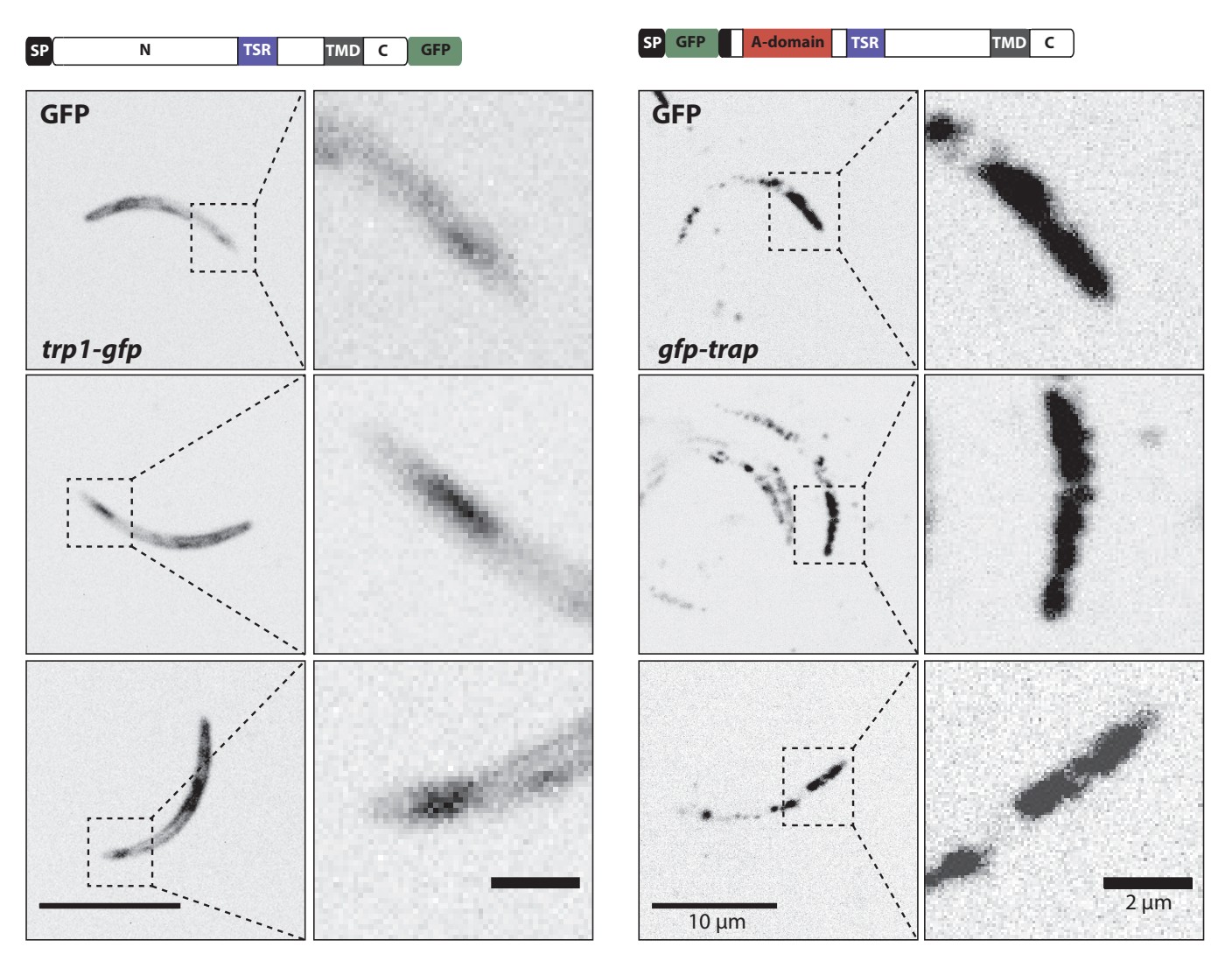

**Figure 8.** Localization of TRP1-GFP and the micronemal protein TRAP. Comparison of salivary gland sporozoites expressing C-terminally tagged TRP1 (*trp1-gfp*) with salivary gland sporozoites expressing N-terminally tagged TRAP (*gfp-trap*). Zoomed images all show the apical tip of the sporozoites. Three different sporozoites are displayed for each strain. While TRAP shows a micronemal localization, predominantly at the apex of the sporozoite, TRP1 localizes close to the plasma membrane and accumulates at the rear end of the sporozoite.

wild-type oocysts showed no differences in the appearance of the oocyst wall. Even 24 days post infection, *trp1(-)mCh* oocysts showed little difference in morphology and no difference in oocyst wall thickness when compared to wild-type oocysts 12 days post infection (*Figure 10—figure supplement 1*). These observations could suggest that proteolytic degradation of the oocyst wall is not affected in oocysts lacking TRP1. Alternatively, proteolytic degradation might happen only shortly before or during egress and is therefore hard to observe by electron microscopy in wild-type oocysts. Clearly, in vivo imaging will be needed to confirm whether the different types of egress also occur in living mosquitoes. Nevertheless, our assays will allow a more quantitative description of sporozoite egress from oocysts as already shown in *Figure 9* with the comparison between *wt*, *trp1 (-)* and *sera5(-)* parasites.

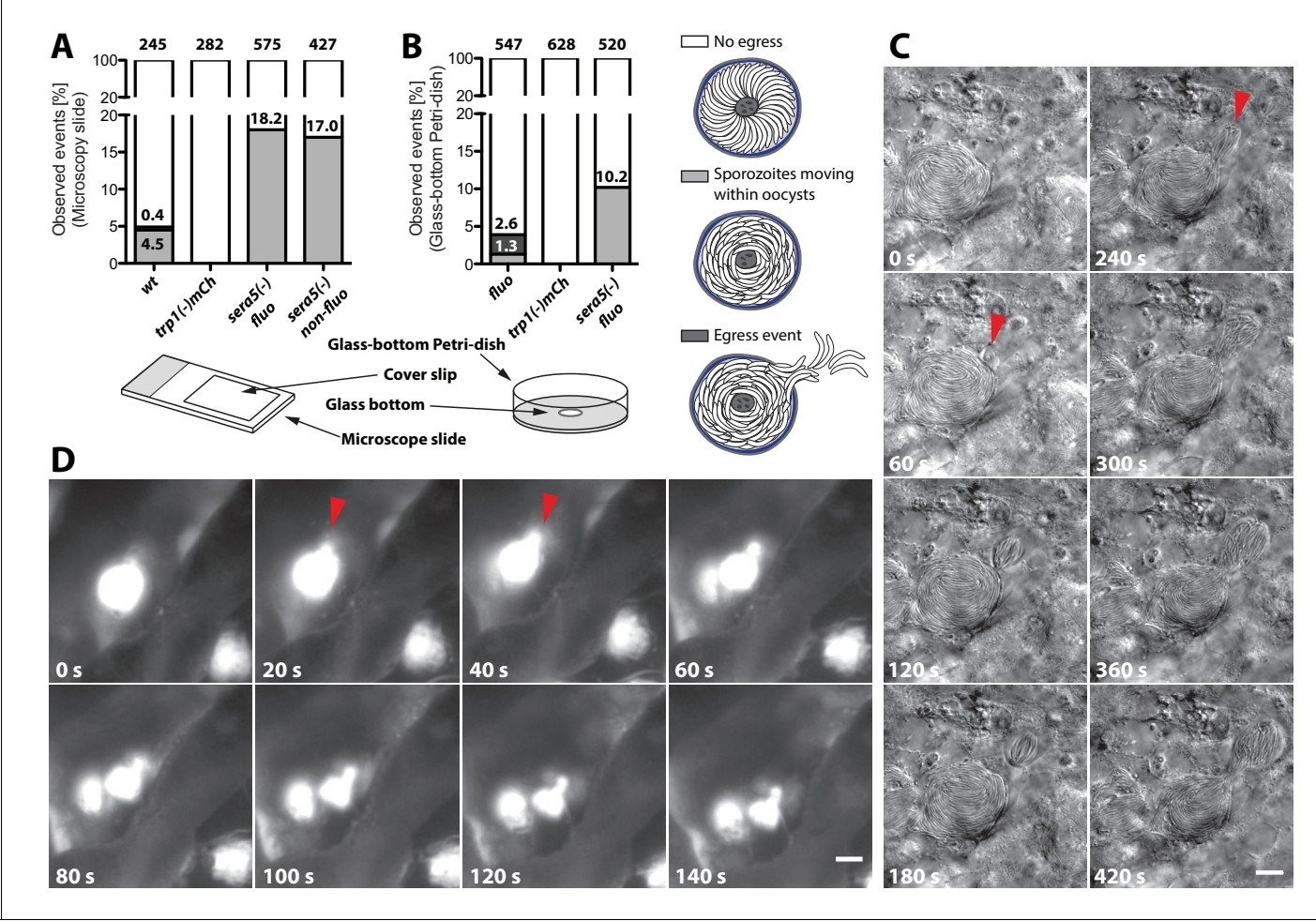

**Figure 9.** *trp1(-)* sporozoites do not egress from oocysts and do not show intra-oocyst motility. (A) Distribution of egress events (dark grey) and oocysts containing motile sporozoites (light grey) in control (*fluo*), wild-type (*wt*) or *sera5(-)* and *trp1(-)mCh* oocysts on a microscope slide covered with a cover slip or (B) uncovered in a glass-bottom Petri-dish. As control for a non-egressing strain, a fluorescent (*sera5(-) fluo*) and a non-fluorescent (*sera5(-) non-fluo*) SERA5 knockout line were tested. The different sample preparation methods are depicted below the graphs. Sporozoites budding from oocysts in a sporosome-like manner as well as spontanous bursting of oocysts were classified as egress events (*Videos 4–6*). (C) Time lapse of a budding event under a cover slip. A wild-type oocyst with budding sporozoites is shown. The start of two budding events is indicated with red arrows. Scale bar: 10 μm. See also *Video 3*. (D) Bursting of an oocyst in a glass-bottom Petri-dish. An oocyst expressing GFP bursting and releasing sporozoites is shown. Scale bar: 20 μm. See also *Video 7*.

## How could TRP1 interact with other proteins to mediate egress?

What role could TRP1 play during sporozoite egress from oocysts? Many TSR-containing proteins in *Plasmodium* have been studied and shown to have functions in gliding motility. The best example is TRAP, which is believed to interact with actin filaments below the plasma membrane to guide motility possibly by direct force transduction to the substrate. TRAP function is crucial as sporozoites that lack TRAP are not able to perform productive motility (*Sultan et al., 1997*) and are probably, as a consequence, also unable to invade the salivary glands and infect the mammalian host. Other proteins of the TRAP-family were shown to have similar functions. CTRP, for example, fulfills the same role as TRAP but in ookinetes (*Dessens et al., 1999*), and S6/TREP-deficient sporozoites have been shown to be less capable of gliding motility and invasion of salivary glands (*Steinbuechel and Matuschewski, 2009*; *Combe et al., 2009*). Interestingly, TRAP-related proteins that lack the penultimate tryptophan in the cytoplasmic tail domain were also shown to support motility. The protein SSP3 is, for example, important for continuous movement of sporozoites (*Harupa et al., 2014*). To

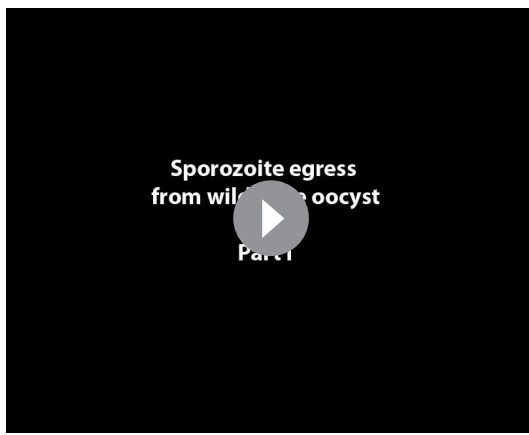

**Video 3.** Sporozoites are budding from a wild-type oocyst. Movie in differential interference contrast (DIC) of wild-type sporozoites moving inside an oocyst and budding from the oocyst in a sporosome. Imaged on an Axiovert 200M (Zeiss) with a 63x (N.A. 1.3) objective. Time between frames: 30 s. These time-laps series were taken subsequently with ~5–10 s between each series.

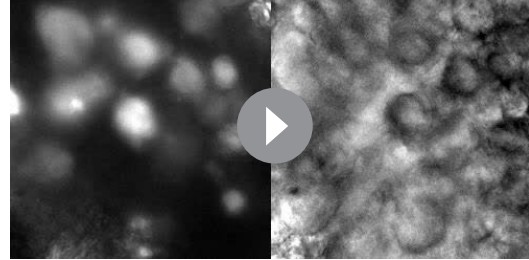

**Video 4.** Oocyst rapidly bursting. Movie showing an oocyst of the *fluo* control line rapidly bursting and disappearing. Fluorescence and differential interference contrast (DIC) are shown beside each other. Imaged on an Axiovert 200M (Zeiss) with a 10x (N.A. 0.5) objective. Time between frames: 30 s.

accomplish these functions, TRAP-family proteins are localized at the plasma membrane. Therefore, it is likely that TRP1 is also at least partially secreted to the parasite surface, where it could initiate signaling from the outside as has been speculated recently to be the case for sporozoite surface proteins (*Kappe et al., 1999*; *Quadt et al., 2016*; *Bane et al., 2016*). Indeed surface proteomics of sporozoites (*Lindner et al., 2013*) showed that TRP1, even if just at low amounts, can be detected on the outside of the sporozoite plasma membrane. Once on the surface, TRP1 could lead, for example, to sustained microneme secretion that would maintain or initiate gliding motility. Alternatively, it might play a role in motility by contributing to the disconnection of TRAP from actin filaments at the rear end, where the actin filament binding protein coronin was recently also shown to be localized (*Bane et al., 2016*). Although these functions remain highly speculative, in the absence of TRP1, there is no motility within oocysts and possibly reduced secretion. A scenario involving TRP1 in signal transduction is not at odds with the observation that isolated *trp1(-)* sporozoites can glide, as other surface proteins can probably also transduce or modulate signals. Also it appears that many diverse ligands can activate sporozoites (*Perschmann et al., 2011*). Beside the direct interaction with ligands that make contact with the sporozoite surface, TRP1 could also assist in the correct trafficking and secretion of other proteins, as shown for MIC2 and the MIC2-associated protein (*Huynh et al., 2003*).

## Putative pathways that could trigger sporozoite egress from oocysts

Clearly, the processes and factors that lead to parasite egress from host cells in general and egress of sporozoites from oocysts in particular are still poorly understood. The factors that are known to date are the serine repeat antigen 5, SERA5 (previously named egress cysteine protease 1; ECP1), the GPI-anchored circumsporozoite protein, CSP (*Wang et al., 2005*; *Aly and Matuschewski, 2005*; *Tewari et al., 2002*), the *Plasmodium* cysteine repeat modular proteins PCRMP3 and PCRMP4 (*Douradinha et al., 2011*) and the LCCL-domain-containing proteins PfCCp2 and PfCCp3 (*Pradel et al., 2004*)

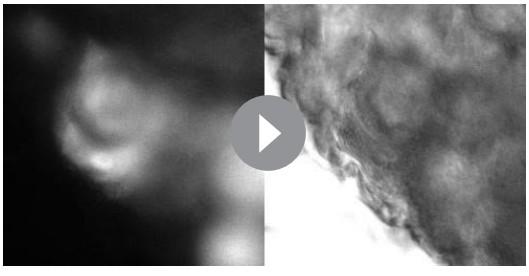

**Video 5.** Sporozoites moving inside oocyst. Movie showing *fluo* control sporozoites moving inside an oocyst. Fluorescence and DIC are shown beside each other. Imaged on an Axiovert 200M (Zeiss) with a 10x (N.A. 0.5) objective. Time between frames: 30 s.

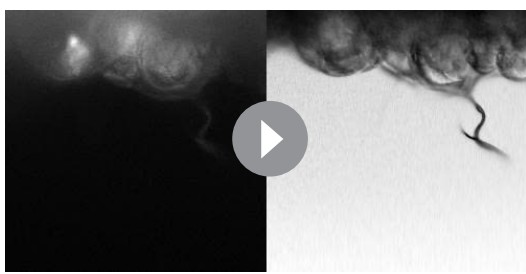

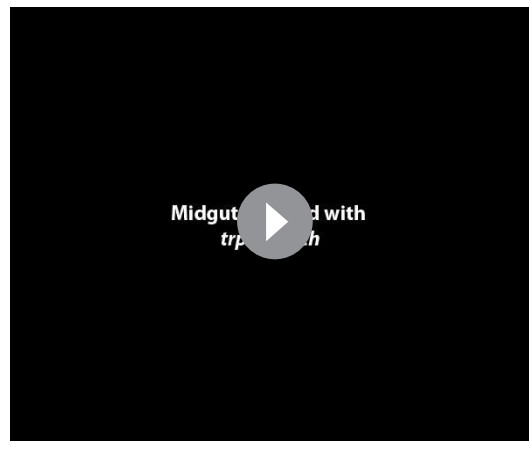

**Video 6.** Sporozoites budding from oocyst. Movie showing *fluo* control sporozoites possibly budding from an oocyst. Fluorescence and DIC are shown beside each other. Imaged on an Axiovert 200M (Zeiss) with a 10x (N.A. 0.5) objective. Time between frames: 30 s.

**Video 7.** Long-term imaging of mosquito midguts infected with *trp1(-)mCh*, *sera5(-) fluo* and the fluorescent reporter line *fluo*. Midguts of mosquitoes infected with *trp1(-)mCh*, *sera5(-) fluo* or a control line were imaged for 1 hr with 30 s per frame.
The video shows, consecutively, a midgut infected with *trp1(-)mCh*, a midgut infected with a fluorescent control line and the *sera5(-) fluo* line. Note the absence of any sporozoite movement in the trp1(-)mCh movie, the bursting of an oocysts in the center of the wt control movie, and the intra-oocyst motility of sporozoites in several oocysts in the *sera5(-) fluo* movie. Imaged on an Axiovert 200M (Zeiss) with a 10x (N.A. 0.5) objective. Time between frames: 30 s.

(*Table 1*). Sporozoites lacking SERA5, PCRMP3, PCRMP4, PfCCp2 and PfCCp3 fail to egress from oocysts (*Aly and Matuschewski, 2005*; *Douradinha et al., 2011*; *Pradel et al., 2004*), whereas parasites lacking CSP do not complete sporozoite formation (*Ménard et al., 1997*). The change of four basic amino acids into alanines within region II+ of CSP allowed normal sporozoite development but blocked egress from oocysts (*Wang et al., 2005*). All six mutant parasites lines retained their capacity to migrate actively once sporozoites were mechanically released from oocysts (*Table 1*). Another protein, the sporozoite invasion associated protein-1 (SIAP-1), possesses no recognizable structural or functional domains but has been identified as having a partial role in sporozoite egress. Parasites that lack SIAP-1 have a strongly reduced rate of sporozoites egressing from oocysts but are still able to enter into salivary glands in low numbers (*Engelmann et al., 2009*). In contrast to parasite lines that lack the previously mentioned proteins (SERA5, PCRMP 3 and 4, PfCCp 2 and 3) or parasite lines that contain specific mutations within CSP, SIAP-1 knockout sporozoites also fail to undergo efficient gliding motility (*Engelmann et al., 2009*). Similarly, the GPI-anchored micronemal antigen GAMA (previously named PSOP9) has been shown to be essential for egress from oocysts. Sporozoites that lack GAMA are also not able to perform gliding motility in vitro and within oocysts ex vivo (*Ecker et al., 2008*). Interestingly, GAMA is expressed in all *Plasmodium* stages that contain micronemes and has been discussed as a potential vaccine candidate against blood stages (*Hinds et al., 2009*; *Arumugam et al., 2011*). This ubiquitous expression profile makes it unlikely that GAMA has a specific function in sporozoite egress from oocysts but suggests that it is important for egress and invasion in general. Therefore the egress defect of *gama(-)* sporozoites presumably causes a defect within the micronemes that has more or less impact dependent on the observed parasite stage. This hypothesis is supported by the observation that *gama(-)* ookinetes produce 78% fewer oocysts than wild-type ookinetes, indicating that the loss of GAMA has an effect prior to oocyst development.

We speculate that two different pathways, involving either intracellular or extracellular signals, might trigger the release of sporozoites from their oocyst. The intracellular pathway might require quorum sensing between the sporozoites of an oocyst that reponds to the presence or absence of a specific factor once sporulation is completed. This signal might be followed by secretion of micronemal and possibly exonemal proteins (e.g. SERA5) that induce both sporozoite motility and degradation of the oocyst wall. While continuous motility as well as inflow of extracellular factors might further enhance protein secretion, the oocyst envelope might become more and more fragile,

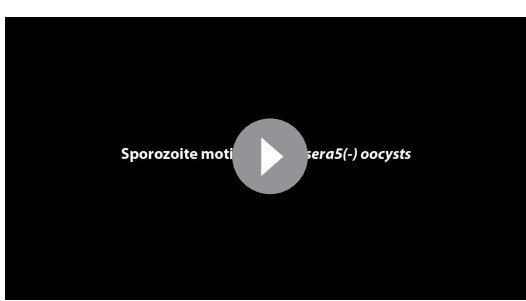

**Video 8.** *sera5(-) fluo* sporozoites moving inside an oocyst. Movie in differential interference contrast (DIC) showing intra-oocyst motility of *sera5(-) fluo* sporozoites within an oocyst. Imaged on an Axiovert 200M (Zeiss) with a 63x (N.A. 1.3) objective. Time between frames: 1 s.

leading eventually to sporozoite egress. The extracellular pathway might be initiated by the permeabilization of the oocyst wall, either by intracellular factors that are secreted after sporulation or by the breakdown of the oocyst wall that is not longer maintained once sporulation is completed. The inflow of extracellular factors might initiate the secretion process, which leads, as described for the intracellular pathway, to increased motility as well as breakdown of the oocyst wall and finally sporozoite egress. Independent of the pathway that triggers sporozoite egress, degradation of the oocyst wall is probably an interplay of multiple factors that do not lead to a uniform breakdown of the oocyst envelope but create focal weakenings that enable local release of sporozoites or budding of sporosomes (*Figure 10*).

The case for SERA5 and CSP appears clearest in this scenario as the lack of the protease (SERA5) has no influence on gliding and neither has the change of the four amino acids in CSP (*Wang et al., 2005*; *Aly and Matuschewski, 2005*). While clearly speculative, this first model of a cascade of possible events linking secretion, motility and proteolysis can be challenged with future work that should aim to identify missing factors and to dissect their functional role and interplay in sporozoites' egress from oocysts. To this end, a combination of double-knockouts and our imaging assays will constitute crucial tools.

## TRP1 is probably trafficked through the ER

Parasites that lack proteins involved in gliding motility (TRAP and CP$\beta$) do not enter into salivary glands (*Sultan et al., 1997*; *Ganter et al., 2009*) and decreased gliding motility (S6/TREP/UOS3, Coronin, CSP and PAT) often goes along with decreased salivary gland invasion (*Coppi et al., 2011*; *Kehrer et al., 2016b*; *Steinbuechel and Matuschewski, 2009*; *Combe et al., 2009*; *Bane et al., 2016*; *Tewari et al., 2002*; *Mikolajczak et al., 2008*). Hence parasites that lack proteins involved in both egress and motility are also impaired in salivary gland invasion and thus these proteins are essential for life cycle progression at two subsequent steps. In this context, TRP1 plays a unique role as it mediates egress but appears to have no role in gliding. Therefore, we assumed that sporozoites that lack a functional TRP1 would enter into salivary glands. Interestingly, we could detect hemolymph sporozoites in low numbers, especially at later time points in mosquitoes highly infected with *trp1(-)mCh* parasites. We assume that in these mosquitoes, some of the oocysts released sporozoites, probably due to mechanical stress or oocyst wall degradation due to an arrest in oocyst wall formation. The numbers of hemolymph sporozoites were with a few hundred to a few thousand sporozoites, high enough to expect at least a few hundred sporozoites in the salivary glands. However, this was never observed.

Observations similar to those made for the knockout lines *trp1(-)* and *trp1(-)mCh* were also made for the strains *gfp-trp1ΔN* and *gfp-trp1ΔC*. *gfp-trp1ΔC* parasites tended to resemble the phenotype of both knockout lines, whereas *gfp-trp1ΔN* parasites showed almost normal egress from oocysts but were not able to invade the salivary glands. These deficiencies can be explained differentially. The *gfp-trp1ΔC* mutant lacks both the cytoplasmic tail domain (CTD) and the putative micronemal targeting signal F/Y/WXXΦ (*Di Cristina et al., 2000*; *Bhanot et al., 2003*), which consists of an aromatic amino acid on position one and an hydrophobic amino acid (Φ) on position four. Micronemal proteins that lack this signal are directly targeted to the plasma membrane and secreted (*Di Cristina et al., 2000*). However, we are not able to distinguish if this is also true for *gfp-trp1ΔC* parasites because we could not observe any GFP expression in this line. Moreover, we cannot exclude the possibility that interactions with other proteins at the CTD are needed for TRP1 to be functional. Curiously, parasites carrying the full-length C-terminally tagged TRP1, albeit processed, showed no defect in their capacity to exit the oocysts, thus suggesting that an C-terminal GFP does not impair

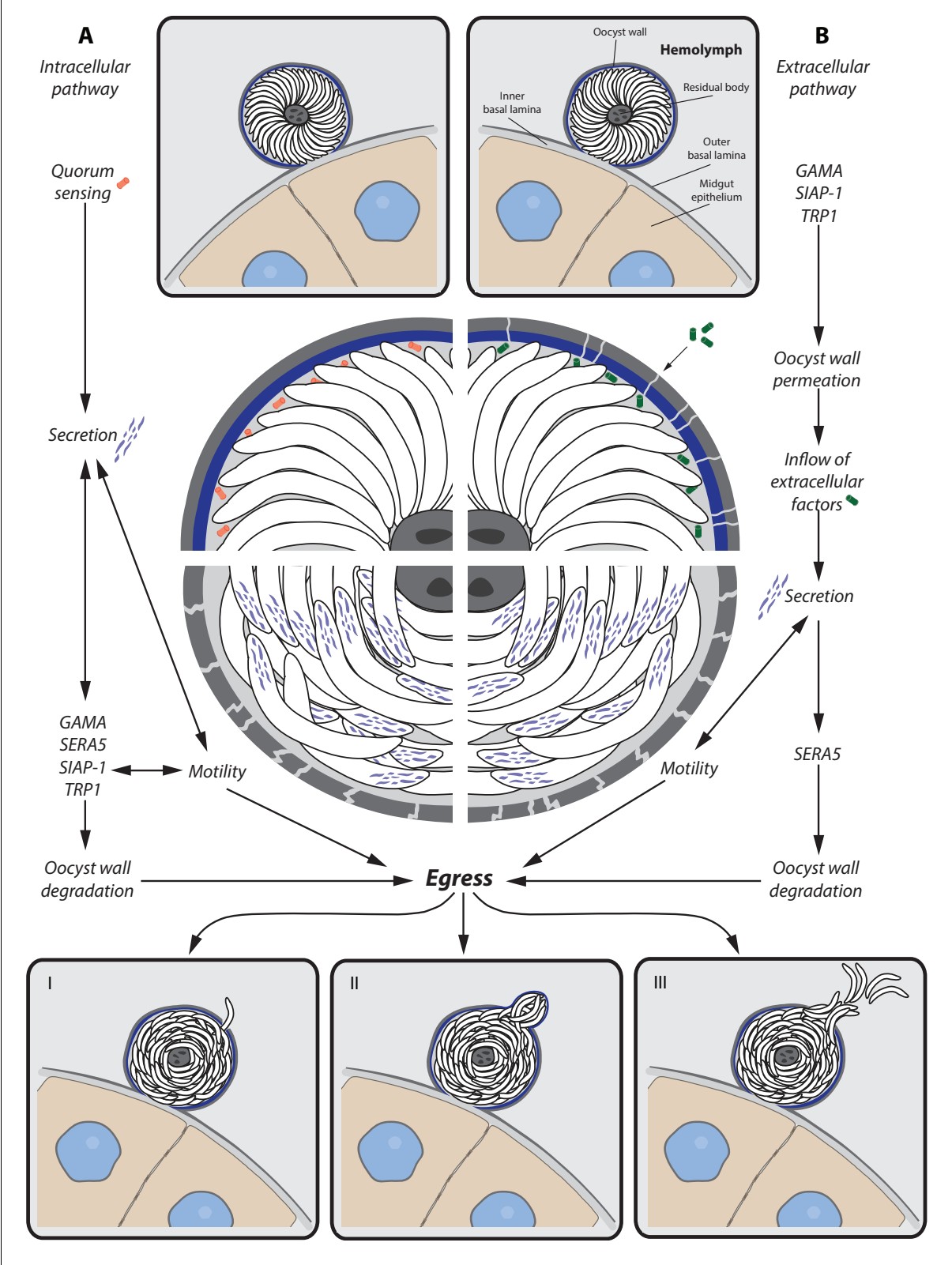

**Figure 10.** Potential model for sporozoite egress from oocysts. Hypothetical model of a cascade of events that lead to sporozoite egress from oocysts. (A) Intracellular pathway — possible quorum sensing between sporozoites leads to secretion of proteins (e.g. GAMA, SERA5, SIAP-1, TRP1) that trigger gliding motility and degradation of the oocyst wall, which is followed by sporozoite egress. (B) Extracellular pathway — expression of factors (e.g. GAMA, TRP1, SIAP-1) leads to permeabilization of the oocyst wall and inflow of extracellular factors. Inflowing factors trigger secretion of proteins that

*Figure 10 continued on next page*

*Figure 10 continued*

not only activate gliding motility but also degrade the oocyst wall (e.g. SERA5), which is followed by sporozoite egress. Egress of sporozoites can occur in different ways. (I) Single sporozoite egress — sporozoites migrate through thin holes in the oocyst envelope. (II) Sporosome formation — many sporozoites stretch the oocyst wall, leading to the formation of sporozoite filled vesicles (sporosomes) that bud from the oocyst. (III) Bursting oocyst — rapid rupture of the oocyst wall.

The following figure supplement is available for figure 10:

**Figure supplement 1.** Electron microscopy of control (*fluo*) and *trp1(-)mCh* oocysts.

this TRP1 function. However, it did not completely complement salivary gland invasion (*Table 2*). This suggests that the C-terminus might function differently in oocyst exit and salivary gland entry.

In the *gfp-trp1ΔN* mutant, the complete N-terminus of TRP1 is missing, which had a crucial effect on TRP1 function. Although *gfp-trp1ΔN* sporozoites could not be detected in the salivary glands and were not transmitted by the bites of infected mosquitoes, this line showed an egress rate that was only slightly decreased compared to wild-type. This suggests that the C-terminal part of the protein is required for sporozoite egress from oocysts, whereas both C- and N-terminus are needed for salivary gland invasion. Interestingly, *gfp-trp1ΔN* was the only N-terminally tagged parasite line that was fluorescent, and the GFP-TRP1ΔN fusion protein could be visualized in oocysts and free sporozoites (*Figures 6* and *7*). The tagged protein localized to membranes within the parasites, especially around the nucleus, which we believe is the location of the endoplasmatic reticulum (ER). This indicates that TRP1 has a signal peptide and enters the secretory pathway similar to micronemal proteins. However, a distinct GFP signal could be observed only in *gfp-trp1ΔN* parasites but not in *gfp-trp1ΔC*, *gfp-trp1comp* or *gfp-trp1* parasites. Interestingly, all of these lines transcribe a *gfp:trp1* fusion transcript (*Figure 5C*), indicating that transcription takes place as predicted. This evidence, together with the observation that the N-terminus of TRP1 clearly varies in size and is less conserved between different *Plasmodium* species (*Figure 5A*, *Supplementary file 1*) and the fact that we observed only a very small TRP1-GFP fusion protein by western blotting (*Figure 5D*), suggests that post-translational processing of TRP1 takes place at the N-terminal end, as shown for MIC5 and M2AP (*Brydges et al., 2008*; *Harper et al., 2006*), and probably also closer to the transmembrane domain. Therefore GFP might be cleaved off and rapidly degraded in *gfp-trp1ΔC*, *gfp-trp1* and *gfp-trp1comp* parasites but not in *gfp-trp1ΔN* parasites. Further mutants could be generated to investigate these possibilities and to investigate putative cleavage site(s). The low band detected by western blotting might also suggest that the TSR domain is lost. Whether this loss is functionally required would also constitute an interesting avenue of future research.

In conclusion, by imaging oocyst egress of *Plasmodium* sporozoites at the mosquito midgut wall in situ, we were able to observe a variety of egress events indicating that release of sporozoites from oocysts occurs in different ways. In addition, we identified the TRAP-related protein TRP1 as crucial for sporozoite egress and salivary gland invasion in *Plasmodium berghei*, suggesting hypothetical cascades of events that drive sporozoite egress.

## Materials and methods

### Bioinformatic analysis

*Plasmodium* sequences were retrieved from PlasmoDB (http://plasmodb.org/plasmo/, version 26) and multiple sequence alignments were performed with MUSCLE (http://www.ebi.ac.uk/Tools/msa/muscle/). Potential signal peptides and transmembrane domains were predicted using SignalP (http://www.cbs.dtu.dk/services/SignalP/), SMART (http://smart.embl-heidelberg.de/) and TMHMM (http://www.cbs.dtu.dk/services/TMHMM-2.0/). The pI values of cytoplasmic tail domains (CTDs) were calculated with Expasy (http://web.expasy.org/compute_pi/).

Depiction of the identity and similarity index was based on a multiple sequence alignment with *Pb*TRP1 (PBANKA_0707900), *Pv*TRP1 (PVX_089230), *Pk*TRP1 (PKNH_1316700) and *Pf*TRP1 (PF3D7_0822700). *Pb*TRP1 was used as reference and all parts of sequences that did not align with *Pb*TRP1 were removed. The identity of amino acids was scored manually by determining the number

**Table 4.** Primer sequences. Primers used for the generation of the different parasite lines, genotyping and reverse transcriptase (RT-) PCR.

| Primer no. | Sequence |
| --- | --- |
| P99 | CTAGCTAGCTTAATCATTCTTCTCATATACTTC |
| P232 | CGCGGATCCTTACTTGTACAGCTCGTCCATGC |
| P234 | CTTGCACCGGTTTTTATAAAATTTTTATTTATTTATAAGC |
| P583 | AGTCATGCTGTTTCATGTGATC |
| P600 | CCCAAGCTTCAAAAAAGCAGGCTTGCCG |
| P601 | GCCGATATCCAAGAAAGCTGGGTGGTACCC |
| P606 | GTAGGTCGACTGCTTAAACAGAAATTTCTGAACTTTGTTAGG |
| P607 | GTAGGAATTCATCATGGTTCAGCTTTCATAAAAATCTATATGG |
| P608 | GTAGAAGCTTGAGCTAAATAATAATGACACCGATTTAACGAG |
| P609 | GTAGCTCGAGCATCTACTACTCATAATACACTTAGTGGAAGTACG |
| P610 | GTAGCCGCGGTGCTTAAACAGAAATTTCTGAACTTTGTTAGG |
| P611 | GTAGGACATATGTCTTCCACCTCCACCATTATCGTATTTTTTCAAAGTAGGACCAATCCA |
| P612 | GTAGGGCGCCGGTGGAGGTGGATGGATTGGTCCTACTTTGAAAAAAATACGATAAT |
| P616 | GTAGGGATCCCAAAGCTGAAACTGATGAACCCATAGATG |
| P657 | GGCATTTAAAACTACTATAGGATGTGGG |
| P682 | CTCAAGGGTTTGATCAAGAAACTGCAG |
| P694 | TAACCATCAAAACATCTCGATCTTTCGAG |
| P695 | AATTTCTTTGACAATTAAATAAACAAGATATATCGCTG |
| P698 | AAATGTAATTTTAGTTCTTGGTCAGATTGGTCAG |
| P699 | ATTATCGTATTTTTTCAAAGTAGGACCAATCCA |
| P887 | GAAGAATATAATTCGATACATATGTTTAGACAAAATC |
| P1296 | GCGGGATCCATGAGTAAAGGAGAAGAACTTTTC |
| P1408 | CATTTTCAGATGGTGTTTCAGTTTGTAC |
| P1409 | CATATGAACTACATGCGTTAGAAGC |
| P1410 | GATGATGATGATGATGAAAATAATGACATG |
| P1411 | CACCATCAAAACGTAATGAAGCTG |
| P1444 | CAAATGCCTCCTGACCAGGC |
| P1597 | GTAGCCGCGGGATGGAAGTTCAAATATGTGTAGACTTACCTTATTG |
| P1562 | GTAGGACATATGTCTTCCACCATCTTTCTTTATGGTATCTGTAATTATATCATTTTCAG |
| P1564 | GTAGGTCGACCACTTAAATTTAATGATTAAATGGTGTGTACATTTCTAC |
| P1565 | GTAGGATATCCATATACATAATACACTTATAGACACATTTAAATATG |
| P1566 | GTAGAAGCTTGACATAGTCATCACAATATTCATTATTCATATATCATAC |
| P1567 | GTAGCTCGAGCAATTTTCCCTTTATAATATTCTGTCTCTTTACATTGC |
| P1595 | GTAAATAAGAATATGCATATACATGGGTG |
| P1596 | CTGTTATAGTATGGGCCATGTTTCTG |
| P1602 | CAGAGATCCTGAATACGACCCTAG |
| P1603 | CTTTCTTCTGAAACATTATCCTGTAAGC |

of conserved amino acids at each position. No consensus was scored as 1, consensus in two sequences was scored as 2 and accordingly 3 or 4 if more sequences showed the same amino acid at the same position. The similarity index was determined by the same method but amino acids were grouped according to chemical properties as acidic (D, E), basic (K, R, H), polar (S, T, N, Q) or hydrophobic (A, V, I, L, M, F, Y, W). The amino acids cysteine, glycine and proline were not grouped and were compared individually. The obtained data was smoothed with the running average of 10 and plotted with GraphPad Prism 5.0 (GraphPad, San Diego, CA, USA).

## Generation of parasite lines *trp1(-)*, *trp1(-)rec* and *trp1(-)mCh*

*trp1(-)* parasites were generated by amplifying 825 bp upstream of PBANKA_0707900 via PCR with the primers P606 and P607 (*Table 4*). The product was used as 5' UTR for homologous recombination and cloned in front of the positive-negative selection marker *hdhfr-yfcu* in the Pb262 vector (*Deligianni et al., 2011*). In a second step, the 3' UTR (1,040 bp) was amplified with the primers P608 and P609 and cloned in the Pb262-PBANKA0707900-int vector downstream of the selection cassette to enable double crossover homologous recombination via both introduced sequences (5' and 3' UTR) and therefore exchange of the *trp1* open reading frame (ORF) with the selection cassette. The final vector Pb262-PBANKA0707900-KO was digested (*SalI* and *XhoI*), purified (High Pure PCR Product Purification Kit, Roche) and transfected into the *P. berghei* strain ANKA using standard protocols (*Janse et al., 2006*). Subsequently, parasites that integrated the desired DNA fragment were selected by administration of pyrimethamine (0.07 mg/mL) in the drinking water. *trp1(-)* parasites were then cloned to generate isogenic populations and negatively selected using 5-fluorocytosine (1 mg/mL) (*Lin, 2011*) to give rise to *trp1(-)rec* parasites that lost the selection cassette (*Figure 2—figure supplement 1*). *trp1(-)rec* parasites were cloned and used for complementation approaches with *gfp*-tagged full-length (*gfp-trp1comp*) and N- and C-terminally (*gfp-trp1ΔN* and *gfp-trp1ΔC*) truncated *trp1* constructs (*Figure 4—figure supplement 1*). The *gfp*-tagged full-length *trp1* construct was also transfected into the *P. berghei* ANKA strain to generate the parasite line *gfp-trp1*, which is genetically identical to *gfp-trp1comp*. In addition to *trp1(-)*, a second knockout line was generated to track promoter activity of *trp1* in vivo as follows. To generate the promoter-reporter construct, the 5'UTR of *trp1* was amplified with the primers P606 and P887 (858 bp). The construct is based on the Pb262 vector that used the same selection cassette as before but which also contained the *mCherry* gene followed by a *dhfs* terminator. The generated 5' UTR was cloned via *SalI* and *NdeI* directly in front of the *mCherry* gene to enable transcription in vivo upon *trp1* promoter activation (*Figure 2—figure supplement 1*). The 3'UTR was amplified and cloned as described above to enable double crossover homologous recombination. The final vector was transfected into the *P. berghei* ANKA strain as described above. Note that the distance between the *trp1* ORF and its neighboring downstream gene (PBANKA_0708000; SEC23) amounts to just 291 bp. To avoid an influence in transcription of PBANKA_0708000 but ensure efficient recombination, we decided to leave a part of the *trp1* coding sequence attached to the 3' UTR. Therefore, all generated knockout lines described in this study still contain 609 bp of the *trp1* ORF but will be referred to as *trp1* knockout.

## Generation of *sera5(-) fluo* and *sera5(-) non-fluo*

The fluorescent and non-fluorescent *sera5(-)* lines were generated in a similar way as the *trp1(-)* and the *trp1(-)mCh* line. The 5'UTR (1,081 bp) of *sera5* was amplified with the primers P1564 and P1565 and ligated by *SalI* and *EcoRV* in the Pb262 vector. In a next step, the 3'UTR (1,012 bp) of *sera5* was amplified with the primers P1566 and P1567 and ligated by *HindIII* and *XhoI* in the Pb262 vector that contained already the *sera5* 5'UTR. The final vector was digested and purified as described previously. As the designed construct contained no additional fluorescent marker within the integrated sequence, transfection was performed in the fluorescent background line *fluo* and in *wt* to generate a fluorescent and a non-fluorescent *sera5(-)* strain.

## Generation of the parasite lines *gfp-trp1comp*, *gfp-trp1*, *gfp-trp1ΔN*, *gfp-trp1ΔC* and *trp1-gfp*

Complementation of TRP1-knockout parasites was achieved with three different constructs encoding either full-length *trp1* or N- and C-terminal truncated mutants (*Figure 4—figure supplement 1*). We

used the Pb238 vector (*Deligianni et al., 2011*) as template for all three complementation constructs. The vector contains the positive selection marker human *dhfr* controlled by the *ef1α* promoter and a *dhfr* terminator from *P. berghei*, as well as a *gfp* upstream of the selection cassette. The 5'UTR, which included the sequence encoding the signal peptide of *trp1* (989 bp), was amplified with the primers P610 and P611 and fused (by *SacII* and *PshAI*) with the *gfp* gene to tag *trp1* N-terminally. In a next step, the 3'UTR of *trp1* was amplified with the primers P608/P609 and cloned downstream of the selection cassette to enable integration by double crossover homologous recombination. This vector was named Pb238-PBANKA0707900-int. To generate the vector for complementation with full-length *trp1*, the coding sequence beginning after the signal peptide and the 3'UTR of *trp1* (3,643 bp) were amplified with the primers P612 and P616. The PCR product was subcloned in the pGEM-T-Easy vector (pGEM-TRP1complete) and fully sequenced. Afterwards, the sequence was cloned (using *KasI* and *BamHI*) in the Pb238-PBANKA0707900-int vector downstream of the *gfp* gene to generate the final construct for complementation. To create truncated mutants, the N- and C-terminus in the pGEM-TRP1complete vector was deleted by site-directed mutagenesis with the primers P694/P695 (C-terminus) and P698/P699 (N-terminus). The PCR products were cloned into the Pb238-PBANKA0707900-int vector as described before. To tag TRP1 C-terminally with GFP, the C-terminal end (1,030 bp) of the *trp1* gene was amplified with the primers P1562 and P1597. The Pb238-PBANKA0707900-int vector was digested with *SacII* and *NdeI* to ligate the previously amplified PCR product in front of the *gfp* and to generate a fusion between both genes. All N-terminally tagged constructs were digested (by *SacII* and *XhoI*), purified (High Pure PCR Product Purification Kit, Roche) and transfected into *trp1(-)rec* parasites using standard protocols. The C-terminally tagged construct was digested and purified in the same way but was transfected into *wt* parasites. Subsequently, parasites that integrated the desired DNA fragment were selected via pyrimethamine as described above.

## Generation of isogenic parasite lines

Isogenic parasite lines were generated by serial dilution of parasites obtained from transfections (parental population). Using this method, only single blood stages were injected into 6–10 NMRI mice. Infected mice were bled once parasitemia reached 1–2%. The blood of infected mice was collected and parasites were either frozen as stocks or purified to isolate genomic DNA with the Blood and Tissue Kit (Qiagen Ltd) (*Klug et al., 2016*).

## Mosquito infection

For mosquito infections, mice were infected by intraperitoneal injection of frozen stocks (150–200 µL). Stocks were either completely injected or split for injection into two mice and parasites allowed to grow for 3–5 days. Infected mice were either directly fed to mosquitoes or bled and used for a fresh blood transfer of 20,000,000 parasites into two naïve mice. Parasites in mice that received infected blood were allowed to grow for further 3–4 days. To determine the number of male gametocytes, a drop of tail blood was placed on a microscope slide and incubated at room temperature for 10–12 min. If 1–2 exflagellation events per counting field (40x magnification) were observed, mice were anesthetized and fed to mosquitoes (*Klug et al., 2016*).

## Analysis of oocyst and sporozoite development

To observe the development of oocysts, midguts of 20–30 mosquitoes were isolated at day 12 and day 22 post infection. Midguts were dissected in phosphate buffered saline (PBS) and fluorescent oocysts were manually counted using a stereomicroscope (SMZ1000, Nikon). To investigate the percentage of sporulated and unsporulated oocysts, midguts were also imaged at both time points. Dissected midguts were mounted with a drop of PBS on a microscopy slide and covered with a cover slip. Samples were sealed with paraffin and imaged with an Axiovert 200M (Zeiss) fluorescence microscope using a 63x (N.A. 1.3) objective. At each time point, images of 50–180 oocysts were taken and classified into oocysts that were in the process of budding or contained already mature sporozoites and oocysts that were pre-mature and didn't contain any sporozoites. Experiments were performed in triplicate with different mosquito feedings. Sporozoites were isolated from the midguts, hemolymph and salivary glands of infected mosquitos at days 14, 17/18, 20 and 22 post infection. For each time point, midguts and salivary glands from at least 10 mosquitoes were

dissected in PBS, crushed to release sporozoites and counted using a Neubauer counting chamber. To isolate hemolymph sporozoites, mosquitoes were cooled on ice and the last segment of the abdomen was cut with a syringe. Prepared mosquitoes were flushed by inserting a long drawn Pasteur pipette into the lateral side of the thorax and injected with RPMI (supplemented with 50,000 units/L penicillin and 50 mg/L streptomycin). The hemolymph was thus drained from the abdomen, collected on a piece of foil and transferred to a plastic reaction tube (Eppendorf). Hemolymph sporozoites were counted as previously described for midgut and salivary gland sporozoites. To image protein localization or the expression of fluorescent markers, infected salivary glands and midguts were dissected in RPMI and mounted on a microscope slide. Samples were sealed with paraffin and imaged with a spinning disc confocal microscope (Nikon Ti series) using a 60x objective (CFI Apo TIRF 60x H; NA 1.49).

## Long-term imaging of infected midguts

Oocysts were either imaged on microscope slides or in glass-bottom Petri-dishes (MatTek corporation, USA). For imaging on microscope slides, infected midguts of *trp1(-)mCh* and *wt anka* day 20 to 22 post infection were dissected in Grace's medium (Gibco) and mounted on a microscope slide. Samples were covered with a cover slip, sealed with paraffin and screened for sporozoite movement within oocysts and egress events using an Axiovert 200M (Zeiss) fluorescence microscope with 63x magnification. Imaging in glass-bottom Petri-dishes was performed in the same way, but dishes were filled 10–15 min prior to dissection with 200 µL Grace's medium to allow adjustment to room temperature. Microscopy was performed with 10x magnification, therefore only the fluorescent knockout *trp1(-)mCh* and a fluorescent control line expressing mCherry under the *CSP* and eGFP under the *ef1α* promoter (*fluo*) were used. Two to three midguts were imaged (1 frame every 30 s for 30 min to 1 hr) per line each day from day 12 to day 19 post infection.

## Sporozoite gliding motility assays

To perform sporozoite gliding motility assays, the midguts of 20–30 mosquitoes were dissected in 50 µL RPMI, smashed with a pestle and purified with 17% accudenz as described previously (*Kennedy et al., 2012*). Afterwards, 100 µL of purified sporozoites were mixed with 100 µL of RPMI containing 6% bovine serum albumin (BSA) (ROTH Ltd) and transferred into a 96-well plate with an optical bottom (Thermo Fisher Scientific, Nunc). Hemolymph sporozoites were isolated as described above and centrifuged for 5 min at 7,000 rpm (Thermo Fisher Scientific, Biofuge primo). The excess of supernatant was discarded, sporozoites were resuspended in 100 µL of RPMI and mixed in a 96-well plate with 100 µL of RPMI containing 6% BSA. Plates were spun for 3 min at 800 rpm (Heraeus Multifuge S1) and directly imaged using an Axiovert 200M (Zeiss) fluorescence microscope. Movies were recorded in differential interference contrast (DIC) with 25x magnification and one frame every 3 s. Analysis of movies was performed with the software ImageJ. Only sporozoites gliding in a circular manner for at least one circle over the time of five minutes were classified as moving. All other motility patterns, such as floating, attached, patch gliding, waving, twitching (*Hegge et al., 2009*), were classified as non-moving.

## Infection by mosquito bites and sporozoite injections

To determine the capacity of the generated parasite strains to undergo transmission from vector to host we performed transmission experiments with infected mosquitoes and sporozoite injections. Infected mosquitoes 17 days post infection were separated in the morning in cups of 10 each and starved for 6–8 hr. Four C57Bl/6 mice per experiment were anaesthetized using a mixture of ketamine and xylazine (87.5 mg/kg ketamine and 12.5 mg/kg xylazine). One anaesthetized mouse was put on each cup and mosquitoes were allowed to bite on the ventral side for approximately 20 min. Directly after the feeding or at the latest the next day, mosquitos that had taken a blood meal were dissected to determine sporozoite numbers within midguts. Salivary glands were not dissected because mosquitoes infected with *trp1(-)* or *trp1(-)mCh* never contained salivary gland sporozoites. For sporozoite injections, midgut (MG) sporozoites were dissected from mosquitoes between day 12 to day 16 post infection. Infected midguts were stored in RPMI medium (containing 50,000 units/L penicillin and 50 mg/L streptomycin) and crushed with a pestle to release the sporozoites. Sporozoites were counted in a Neubauer counting chamber and afterwards diluted to the desired

concentration per 100 μL (400,000–500,000 MG sporozoites). For each experiment, four C57Bl/6 mice were injected intravenously. Bitten and injected mice were probed for parasitemia from day 3 to day 10 after injection or exposure to mosquitoes. In addition, the survival of mice was monitored for 30 days post infection. The blood smears were stained in Giemsa solution (Merck) and counted using a light microscope (Zeiss) with a counting grid. The time difference between infection and observation of the first parasite within a blood smear was calculated as prepatency.

## Immunofluorescence on midgut sporozoites

Infected midguts were dissected in PBS in a plastic reaction tube (Eppendorf). Sporozoites were mechanically released with a pestle and fixed in 4% paraformaldehyde (PFA) solution (diluted in PBS) overnight at 4°C. Samples were washed three times with PBS and sporozoites pelleted by centrifugation for 3 min at 10,000 rpm (Thermo Fisher Scientific, Biofuge primo). Sporozoites were blocked (PBS containing 2% BSA) or blocked and pemeabilized (PBS containing 2% BSA and 0.5% Triton-X-100) over night at 4°C. Processed sporozoites were pelleted by centrifugation for 3 min at 10,000 rpm (ThermoFisher Scientific, Biofuge primo) and the supernatant discarded. Samples were incubated with primary antibody solutions (anti-CSP mAb 3D11 (*Yoshida et al., 1980*), 1:5 (cell culture supernatant); rabbit anti-GFP ABfinity, 1:200 diluted) for 1 hr at room temperature (RT) in the dark and subsequently washed three times with PBS. After the last washing step, samples were resuspended in secondary antibody solutions (Cy5 goat anti-mouse, 1:500; AlexaFluor 488 goat anti-rabbit, 1:500 diluted) and again incubated for 1 hr at RT in the dark. Afterwards, samples were washed again three times in PBS, the supernatant discarded and pellets resuspended in 50 μL PBS. Samples were carefully pipetted on microscopy slides and allowed to settle for 10–15 min at RT. Before the solution dried completely, samples were covered with cover slips and 7 μL of mounting medium (ThermoFisher Scientific, ProLong Gold Antifade Reagent). Samples were allowed to set overnight at RT and then kept at 4°C or directly examined with a spinning disc confocal microscope (Nikon Ti series). All images were acquired with 60-fold magnification (CFI Apo TIRF 60x H; NA 1.49).

## Electron microscopy

Infected midguts were dissected as described previously and directly fixed in 2% glutaraldehyde and 2% paraformaldehyde diluted in 0.1 M sodium cacodylate buffer at 4°C overnight. Prepared midguts were washed three times for 5 min in 0.1 M sodium cacodylate buffer and postfixed with 1% osmium (in 0.1 M sodium cacodylate buffer) for 60 min at RT. Samples were again washed two times each with 0.1 M sodium cacodylate buffer and ddH$_2$O. For better contrast, samples were incubated in 1% uranyl acetate (in ddH$_2$O) at 4°C overnight and washed two times for 10 min with ddH$_2$O. Prior to imaging, samples were serially dehydrated with acetone and adapted to embedding solution 'Spurr' (23.6% epoxycyclohexylmethyl-3,4epoxycyclohexyl-carboxylate (ERL); 14.2% ERL-4206 plasticizer; 61.3% nonenylsuccinic anhydride; 0.9% dimethylethanolamine). Treated midguts were embedded in a capsule mould using 'Spurr' and incubated overnight at 60°C. Embedded midguts were sectioned the next day and images were acquired on a JEOL JEM-1400 electron microscope at 80 kV using the TempCam F416 (Tietz Video and Image Processing Systems GmBH, Gautig).

## Western blot

To estimate the molecular weight of the expressed TRP1 and to prove the correct C-terminal tagging with GFP, midguts infected with *csgfp* and *trp1-gfp* parasites were dissected in RPMI medium (containing 50,000 units/L penicillin and 50 mg/L streptomycin). Midguts were smashed with a pestle and free midgut sporozoites purified with accudenz (*Kennedy et al., 2012*). After purification, 100,000 midgut sporozoites per tube were centrifuged for 10 min at 13,000 rpm (Thermo Fisher Scientific, Biofuge primo). The supernatant was discarded and the pellet was lysed in 30 μL RIPA buffer (50 mM Tris pH 8, 1% NP40, 0.5% sodium dexoycholate, 0.1% SDS, 150 mM NaCl, 2 mM EDTA) for ≥1 hr on ice. Probes were mixed with Laemmli buffer (containing 10% β-mercaptoethanol) and denatured for 10 min at 95°C, centrifuged for 1 min at 13,000 rpm (Thermo Fisher Scientific, Biofuge primo) and frozen for 5 min at −20°C. Prepared samples were separated on precast 4–15% SDS-PAGE gels (Mini Protein TGX Gels, Bio-Rad) and blotted on nitrocellulose membranes using the Trans-Blot Turbo Transfer System (Bio-Rad). Membranes were subsequently blocked (PBS containing 0.05% Tween20 and 5% milk powder) and incubated for 1 hr with antibodies directed against GFP

(mouse monoclonal antibody, clones 7.1 and 13.1, Roche, 1:1,000 diluted) or CSP (anti-CSP mAb 3D11 (*Yoshida et al., 1980*), cell culture supernatant 1:50 diluted). Membranes were washed three times (PBS with 0.05% Tween20) and secondary anti-mouse antibodies (NXA931, GE Healthcare) conjugated to horse peroxidase were applied subsequently for 1 hr (1:10,000 dilution). Prior to incubation with antibodies against CSP, which was used as a loading control, blots were treated with mild stripping buffer according to abcam protocols. Signals were detected using SuperSignal West Pico Chemiluminescent Substrate or SuperSignal West Femto Maximum Sensitivity Substrate (Themo Fisher Scientific).

### Reverse transcriptase (RT)-PCR

Total RNA of midgut sporozoites was isolated using the Qiazol reagent according to the manufacturer's protocol (ThemoFisher Scientific). For each strain, 1–2.5 million sporozoites were dissected between day 12 and day 20 post infection and used for RNA isolation. cDNA was prepared from isolated RNA using the First Strand cDNA synthesis kit (ThemoFisher Scientific). Primers used for PCR amplification of *trp1* and *gfp* using Taq polymerase are given in *Table 4*.

### Ethics statement

All animal experiments were performed according to the FELASA and GV-SOLAS standard guidelines. Animal experiments were approved by the responsible German authorities (Regierungspräsidium Karlsruhe, Tierantrag G-283/14; G-134/14). *Plasmodium* parasites were maintained in NMRI mice that were obtained from JANVIER. The prepatency of infected mice as well as parasite growth were determined with C57Bl/6 mice from Charles River Laboratories. All transfections and genetic modifications were done in the *Plasmodium berghei* ANKA background either directly in the wild-type (*Vincke and Bafort, 1968*) or in wild-type derived strains (e.g. *trp1(-)rec*).

### Statistical analysis

Statistical analyses were performed using GraphPad Prism 5.0 (GraphPad, San Diego, CA, USA). Data sets were either tested with a one-way ANOVA or a Student's T-test. A value of $p < 0.05$ was considered significant.

## Acknowledgements

We thank Miriam Reinig and Christian Sommerauer for rearing *Anopheles stephensi* mosquitoes, and Stefan Lindner and Samantha Ebersoll for technical assistance. We thank also Ross Douglas and Mirko Singer for helpful comments on the manuscript and Stephanie Gold, as well as Stefan Hillmer at the Heidelberg University Electron Microscope Core Facility (EMCF), for their help with the EM component of the work. A big thank you to the Biology of Parasitism (BoP) courses 2015 and 2016 at the Marine Biological Laboratory in Woods Hole for helpful discussions and experiments on oocyst imaging. This work was funded by grants from the European Research Council (ERC StG 281719), the Human Frontier Science Program (RGY0071/2011) and the German Research Foundation (SFB 1129). FF is a member of the Heidelberg University cluster of Excellence *CellNetworks*. DK is a member of the Hartmut Hoffman-Berling International Graduate School (HBIGS).

## Additional information

### Funding

| Funder | Grant reference number | Author |
| --- | --- | --- |
| Deutsche Forschungsgemeinschaft | SFB 1129 | Friedrich Frischknecht |
| Human Frontier Science Program | RGY0071/2011 | Friedrich Frischknecht |
| European Commission | ERC StG 281719 | Friedrich Frischknecht |

The funders had no role in study design, data collection and interpretation, or the decision to submit the work for publication.

## Author contributions

DK, Conception and design, Acquisition of data, Analysis and interpretation of data, Drafting or revising the article; FF, Conception and design, Analysis and interpretation of data, Drafting or revising the article

## Author ORCIDs

Dennis Klug, http://orcid.org/0000-0002-9108-454X
Friedrich Frischknecht, http://orcid.org/0000-0002-8332-6668

## Ethics

Animal experimentation: All animal experiments were performed according to the FELASA and GV-SOLAS standard guidelines. Animal experiments were approved by the responsible German authorities (Regierungspräsidium Karlsruhe, Tierantrag G-283/14; G-134/14). For all mosquito feeding procedures mice were anesthetized with a mixture of ketamine and xylazine to minimize suffering.

## Additional files

### Supplementary files

• Supplementary file 1. Alignment of TRP1 homologues from different *Plasmodium* species. Multiple sequence alignment with TRP1 homologues from *P. berghei*, *P. chabaudi*, *P. yoelii 17X*, *P.vivax*, *P. knowlesi* and *P. falciparum 3D7*. Highly conserved residues are written in white and highlighted in black, mostly conserved residues are highlighted in dark grey and less conserved residues are highlighted in light grey. The N-terminus (not present in *gfp-trp1ΔN*) is marked with a green line, the thrombospondin repeat is indicated in blue, the transmembrane domain is marked in orange and the C-terminus (not present in *gfp-trp1ΔC*) is highlighted in yellow. The red line marks a short sequence of 11 amino acids that is duplicated in the tagged lines *gfp-trp1ΔN, gfp-trp1ΔC, gfp-trp1comp* and *gfp-trp1* before and after the *GFP* to ensure the structural integrity of the protein.

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
