## [Decision Letter]

Thank you for submitting your article "Motility precedes egress of malaria parasites from oocysts" for consideration by *eLife*. Your article has been reviewed by three peer reviewers, and the evaluation has been overseen by Urszula Krzych as the Reviewing Editor and Richard Losick as the Senior Editor. The following individuals involved in review of your submission have agreed to reveal their identity: Catherine Braun-Breton (Reviewer #2).

The reviewers have discussed the reviews with one another and the Reviewing Editor has drafted this decision to help you prepare a revised submission. As you will see, among other things, we require a more precise demonstration of processing and cell surface localization of the Trp1 protein. It will help to know if you are prepared to perform the essential experiments, and if so, if you will be able to provide such analysis within a reasonable period of time.

Summary:

The authors identified thrombospondin-related protein 1 as being crucial for egress from oocysts that develop in the midgut of *Plasmodium* infected mosquitos. The results are based on the analysis of two different knockouts lines along with extensive characterization of sporozoite development, motility, ability to reach salivary glands, and mouse infectivity. The findings suggest that Trp1 is required for motility within oocysts as a prelude to egress and that sporozoites egress from oocysts by several different styles, including release of free sporozoites from a defined point(s) or release of encased sporozoites as a group. Although for the most part the work is thorough and the experiments well executed, there are substantial concerns expressed by the reviewers and these include additional data, expansion of the Discussion as well as some degree of toning down the importance of some aspects of the findings.

Essential revisions:

1) The identification of two egress styles is novel, as is the imaging setup to video capture sporozoites egress from oocysts. However, a caution should be exercised about discussing sporosomes as "structures" rather than oocyst wall deformations due to a thinned wall pushed by motile sporozoites (like the plasma membrane deformation induced by motile intracellular *Listeria*). The fact that the N-term domain of Trp1 complemented the egress defect but not the salivary gland invasion deficiency is an interesting phenotype. A more precise localization of the wild type and mutant Trp1 proteins would help to explain/speculate on Trp1 biological roles. Such a study may involve immunoEM where your lab or a collaborator may be prepared to examine the cell surface disposition of Trp1.

2) The part of the Discussion regarding the new egress type should be rewritten to tone-it-down, unless the authors can document the so called sporosomes as clearly defined structures.

3) There is a lack of convincing evidence that Trp1 is on the sporozoites surface. Showing surface expression would support a model of Trp1 involvement in binding extracellular components or transmitting a signal.

4) Because only the *trp1* construct lacking the N-terminal domain of Trp1 is detected by GFP fluorescence, the authors propose that Trp1 is N-terminally processed. This might indeed be the case but other possibilities such a different intracellular location might also explain this observation: the fluorescence of GFP might indeed be altered and thus not detected in some intracellular compartments. The processing of Trp1 should thus be further documented by western-blot experiments.

5) Grater clarification of the sporozoite counts need to be include; the authors should separate midgut sporozoite counts (high numbers) from hemolymph and salivary glands (low numbers) and present for each localization in the different parasite lines.

6) The authors assume TRP1 traffics through the micronemes, yet no confirmatory data is shown. Could co-stains be performed with a bona-fide microneme protein for reference?

7) Mutants for SERA5, *ccp2/ccp3, pcrmp3/pcrmp4,* and *psop9* all fail to egress yet show in vitro motility, so this aspect of the *trp1* mutant is not novel. If the authors have one or more of the previously described egress mutants in hand (SERA5, *ccp2/ccp3, pcrmp3/pcrmp4*, and *psop9*) then perhaps they can image them in the new assay to provide additional insight/distinctions.

8) The observation that motility precedes sporozoite egress requires a confirmatory data by yet another assay.

9) Please expand your Discussion and comment on other previously identified proteins required for sporozoite egress.

10) Also, please comment how is this mutant phenotype different from other sporozoite egress mutants that are motile when physically released from the oocyst. Although it may be difficult to provide conclusive answers as the mutants have not been imaged directly in this new assay, this point should at least be added to the Discussion.

---

## [Author Response]

*Essential revisions:*

1) The identification of two egress styles is novel, as is the imaging setup to video capture sporozoites egress from oocysts. However, a caution should be exercised about discussing sporosomes as "structures" rather than oocyst wall deformations due to a thinned wall pushed by motile sporozoites (like the plasma membrane deformation induced by motile intracellular Listeria). The fact that the N-term domain of Trp1 complemented the egress defect but not the salivary gland invasion deficiency is an interesting phenotype. A more precise localization of the wild type and mutant Trp1 proteins would help to explain/speculate on Trp1 biological roles. Such a study may involve immunoEM where your lab or a collaborator may be prepared to examine the cell surface disposition of Trp1.

We thank the reviewers for appreciating the key novelties of our paper and their constructive critique. We changed “structures” into “vesicle-like deformations of the cyst wall” (subsection “Synchronous activation of mature sporozoites is crucial for effective egress from oocysts” and subsection “Visualizing sporozoite egress: Bursting and budding within vesicle-like structures”, first paragraph), which is indeed a clearer message. In order to investigate the localization of TRP1 more precisely (and also to understand the processing of the protein better – see below) we generated a new parasite line that expresses a C-terminally GFP-tagged TRP1. Although this did not allow us to investigate directly the surface (the GFP-tag is presumably on the insight of the plasma membrane), we could reveal an intriguing polarized localization to the periphery (likely surface). TRP1 is localized mostly at the rear end of the parasite. Interestingly, another protein with such a clear polarized localization to the rear is the actin filament binding protein coronin (recently published by us in Plos Pathogens). Sporozoites lacking coronin have a problem with gliding motility and salivary gland invasion. As *trp1(-)* sporozoites also fail to enter into salivary glands, these proteins might somehow act in a similar way. We added the new data for TRP1-GFP into Figure 5–Figure 8 and the accompanying text in the Results (subsection “TRP1 is post-translationally processed”) and Discussion (subsection “TRP1 is likely trafficked through the ER”, second and third paragraphs). As to immuno EM we don’t hold the technique to high in esteem for sporozoites for various reasons, mostly because in our view it only has produced nice results for highly expressed proteins (e.g. CSP and TRAP), but TRP1 is clearly not among them. Nevertheless, we tried already early on to raise antibodies against TRP1 but failed in two attempts to get an antibody that was specific enough in Western blotting to be used in either immunofluorescence or EM analysis (see Figure 11 and Figure 12). As the N-terminally tagged GFP is cleaved from the protein, we could not use an anti-GFP antibody to probe surface location as we initially hoped. The questions on where TRP1 is cleaved and by which enzyme clearly requires future research. Our also added data on *sera5(-)* parasites suggest that it is not this protease that processes TRP1 (see below).

Author response image 1.A peptide antibody designed against the TRP1 C- terminus does not recognize TRP1 by immunofluorescence and western blot.(**A**) Immunofluorescence on permeabilized (Triton-X 100) midgut sporozoites of *wt* and *trp1(-)mCh*. The staining with αCSP antibodies, which label the sporozoite surface, was included as control to validate the staining procedure. Note that the immunofluorescence with the C-terminal αTRP1 antibody showed in both strains an equivalent intensity suggesting no specific binding to TRP1. (**B**) Illustration of TRP1, the location and sequence (15 last amino acids) of the peptide used for raising the antibody. (**C**) Western blot with *wt* control and *trp1(-)mCh* midgut sporozoites. The C- terminal αTRP1 antibody recognizes an unknown *Plasmodium* or *Anopheles* protein (lower images). CSP was used as a loading control (upper images).**DOI:**
http://dx.doi.org/10.7554/eLife.19157.035

Author response image 2.A peptide antibody designed against the TRP1 N- terminus does not recognize TRP1 by immunofluorescence and western blot.(**A**) Immunofluorescence on permeabilized (Triton-X 100) midgut sporozoites of *wt* and *trp1(-)mCh*. The staining with αCSP antibodies was included as control to validate the staining procedure. The immunofluorescence signal with the N- terminal αTRP1 was not significantly different from background suggesting non- specific binding. (**B**) Illustration of TRP1, the location and sequence (15 amino acids) of the peptide used for raising the antibody. (**C**) Western blot with *wt* (control) midgut sporozoites. The N-terminal αTRP1 antibody doesn’t recognize any *Plasmodium* protein. Note that the unspecific binding between 150 and 250 kDa corresponds to a shading of the membrane but not a distinct band that got enhanced during the scanning of the western blot. CSP was used as a loading control (on the right side). Both images correspond to the same sample on the same western blot.**DOI:**
http://dx.doi.org/10.7554/eLife.19157.036

2) The part of the Discussion regarding the new egress type should be rewritten to tone-it-down, unless the authors can document the so called sporosomes as clearly defined structures.

We renamed the “structures” into “vesicle-like deformations of the cyst wall” and also added a cautionary sentence that states that in vivoimaging will be needed to test which of the observed events do occur (subsection “Visualizing sporozoite egress: Bursting and budding within vesicle-like structures”, last paragraph). We further added cautionary adjectives such as “apparent” (in the first paragraph of the aforementioned subsection) and stated that this observation was “unexpected” (last paragraph) to hopefully clearly state that this observation is not written in stone.

3) There is a lack of convincing evidence that Trp1 is on the sporozoites surface. Showing surface expression would support a model of Trp1 involvement in binding extracellular components or transmitting a signal.

To address the localization of TRP1 in more detail we generated a new parasite line named *trp1-gfp* that expresses TRP1 fused C-terminally to GFP. Indeed *trp1-gfp* parasites showed a much stronger expression of GFP than all other lines and differed in the localization in oocysts and sporozoites significantly from *gfp-trp1:ΔN* parasites. To show the localization in more detail we included the three new Figure 6–Figure 8. In sporozoites TRP1-GFP localized in a polarized fashion at the periphery, probably to the plasma membrane. However, we can’t tell from the images if TRP1 localizes to the inner membrane complex, within the inner membrane space of at the plasma membrane itself. We are also not able to address this question further, for example by immunofluorescence on unpermeabilized sporozoites, since we did not manage to raise an antibody against TRP1. Nevertheless, we think that the new data, together with recently published proteomics data (Lindner et al., Mol Cell Proteomics 2013) are a strong hint that TRP1 is a surface protein that could be involved in signaling processes. We also compared TRP1-GFP localization with that of GFP-TRAP, which revealed a distinct localization, although the signals partly overlap in the apical part, suggesting that TRP1 might be localized in a subset of micronemes or a different set of secretory vesicles.

4) Because only the trp1 construct lacking the N-terminal domain of Trp1 is detected by GFP fluorescence, the authors propose that Trp1 is N-terminally processed. This might indeed be the case but other possibilities such a different intracellular location might also explain this observation: the fluorescence of GFP might indeed be altered and thus not detected in some intracellular compartments. The processing of Trp1 should thus be further documented by western-blot experiments.

Thanks for this suggestion. We now included a western blot in Figure 5 showing that the molecular weight of TRP1 is significantly smaller than predicted. This suggests that not only the N-terminus is cleaved but also the TSR. Whether this is due to a single or multiple cleavage events should be addressed in future research.

5) Grater clarification of the sporozoite counts need to be include; the authors should separate midgut sporozoite counts (high numbers) from hemolymph and salivary glands (low numbers) and present for each localization in the different parasite lines.

Thanks also for these suggestions. We included Table 2 into the paper, where for all experiments the number of counted sporozoites in midguts, hemolymph and salivary glands are listed. Together with the new generated line *trp1-gfp* we added also additional images of oocysts and midgut sporozoites of the parasite line *gfp-trp1ΔN* shown in the new Figure 5–Figure 8.

6) The authors assume TRP1 traffics through the micronemes, yet no confirmatory data is shown. Could co-stains be performed with a bona-fide microneme protein for reference?

Having generated a new parasite line expressing C-terminally tagged TRP1, we compared TRP1-GFP localization with that of GFP-TRAP (new Figure 8). This revealed a distinct localization, although the signals partly overlap in the apical part, suggesting that TRP1 might be localized in a subset of micronemes or a different set of vesicles. Subsets of micronemes have indeed been found in *T. gondii* tachyzoites, which are much larger and hence amenable to super-resolution imaging (Kremer et al., Plos Path 2013). We also applied two different super-resolution imaging techniques on sporozoites with both Prof. Christoph Cremer (GSDM) and Prof. Stefan Hell (STED), but both did not allow us to distinguish between two neighboring micronemes. We currently aim at establishing proximity biotinylation methods for sporozoites, which will ultimately be helpful of biochemically distinguishing such populations. However, currently we only had success with gametocystes (Kehrer et al., Mol Cell Proteom 2016) and ookinetes (unpublished) but with sporozoites we still struggle with the amount of collected protein needed for analysis by mass spectrometry.

*7) Mutants for SERA5, ccp2/ccp3, pcrmp3/pcrmp4, and psop9 all fail to egress yet show* in vitro *motility, so this aspect of the trp1 mutant is not novel. If the authors have one or more of the previously described egress mutants in hand (SERA5, ccp2/ccp3, pcrmp3/pcrmp4, and psop9) then perhaps they can image them in the new assay to provide additional insight/distinctions.*

To address this comment we generated a fluorescent and a non-fluorescent SERA5 knockout line (*sera5(-) fluo* and *sera5(-) non-fluo*). Both lines were tested in the new assays and the generated data were included in Figure 9. Futhermore the ratio of hemolymph and midgut sporozoites respectively salivary gland and midgut sporozoites were included in Figure 3. The sporozoite counts for both lines are also shown in Table 2. Two videos (Video 7 and Video 8) showing intra-oocyst motility in *sera5(-)* oocysts at different magnifications were added. The Figure 3—figure supplement 1 was included to illustrate the generation of both SERA5 knockout strains. Considering that we could confirm (Aly and Matuschewski J. Ex. Med. 2005) intra-oocyst motility in *sera5(-)* parasites but did not see any such motility in *trp1(-)* parasites also suggests that SERA5 does not cleave TRP1 as in such a case both phenotypes should be equivalent.

8) The observation that motility precedes sporozoite egress requires a confirmatory data by yet another assay.

The best such assay would clearly be in vivoimaging, which we discuss at length to be currently not possible. As a “next-best assay”, however, we did include one more mutant that has the most striking phenotype of all currently available oocyst egress mutants in our analysis: *sera5(-)* sporozoites were confirmed to be motile in oocysts in both our assays but were not able to exit as previously reported (Aly and Matuschewksi, J. Ex. Med. 2005). Importantly we quantified this effect and found about 4-8 times as many oocysts that show intra-oocyst motility compared with *wt* parasites, depending on the assay used (Figure 9). This striking finding suggests strongly that movement precedes motility as now stated at the end of the Results section.

9) Please expand your Discussion and comment on other previously identified proteins required for sporozoite egress.

We expanded the discussion on GAMA (formerly named PSOP-9), which has multiple roles in addition to egress of sporozoites (subsection “Putative pathways that could trigger sporozoite egress from oocysts”, first paragraph). We also discuss every single protein revealed to date as being important in sporozoite egress from oocysts in a concise manner but did not expand further from the previous text. Expanding this part would in our view distract from the main flow of the Discussion and decrease readability. We hope that the reviewers will recognize that our paper includes several different transgenic lines and establishes new assay systems that allow (with the inclusion of the *sera5(-)* parasites) a quantitative comparison of sporozoite egress from oocysts for the first time. This leads for the first time to a suggestion of a cascade of events that might mediate sporozoite egress. We thus decided not to expand the discussion on individual proteins even more, especially as we cannot yet come to a final conclusion on the mechanisms at work.

10) Also, please comment how is this mutant phenotype different from other sporozoite egress mutants that are motile when physically released from the oocyst. Although it may be difficult to provide conclusive answers as the mutants have not been imaged directly in this new assay, this point should at least be added to the Discussion.

We include the statements “However, our assays will allow a more quantitative description of sporozoite egress from oocysts as already shown in Figure 9 with the comparison between *wt, trp1(-)* and *sera5(-)* parasites.” and “To this end a combination of double-knock outs and our imaging assays will provide crucial tools” in the Discussion. These were added, as we think that the generation of double mutant lines using the positive-negative selection marker *hdhfr-yfcu* or *crispr/cas9* in combination with our visual assays will ultimately lead the way forward to understand sporozoite egress from oocysts in more detail.